# Phagocytosis by an HIV antibody is associated with reduced viremia irrespective of enhanced complement lysis

David A. Spencer [1,7], Benjamin S. Goldberg [2], Shilpi Pandey[1], Tracy Ordonez[1], Jérémy Dufloo [3,8], Philip Barnette[1], William F. Sutton[1], Heidi Henderson[1], Rebecca Agnor[4], Lina Gao [4], Timothée Bruel [3,5], Olivier Schwartz [3,5], Nancy L. Haigwood[1,6], Margaret E. Ackerman [2] & Ann J. Hessell [1✉]

Increasingly, antibodies are being used to treat and prevent viral infections. In the context of HIV, efficacy is primarily attributed to dose-dependent neutralization potency and to a lesser extent Fc-mediated effector functions. It remains unclear whether augmenting effector functions of broadly neutralizing antibodies (bNAbs) may improve their clinical potential. Here, we use bNAb 10E8v4 targeting the membrane external proximal region (MPER) to examine the role of antibody-mediated effector and complement (C') activity when administered prophylactically against SHIV challenge in rhesus macaques. With sub-protective dosing, we find a 78–88% reduction in post-acute viremia that is associated with 10E8v4-mediated phagocytosis acting at the time of challenge. Neither plasma nor tissue viremic outcomes in vivo is improved with an Fc-modified variant of 10E8v4 enhanced for C' functions as determined in vitro. These results suggest that effector functions inherent to unmodified 10E8v4 contribute to efficacy against SHIV$_{SF162P3}$ in the absence of plasma neutralizing titers, while C' functions are dispensable in this setting, informing design of bNAb modifications for improving protective efficacy.

[1] Division of Pathobiology & Immunology, Oregon National Primate Research Center, Oregon Health & Science University, Beaverton, OR, USA. [2] Thayer School of Engineering, Dartmouth College, Hanover, NH, USA. [3] Virus & Immunity Unit, Department of Virology, Institut Pasteur, Paris, France. [4] Knight Cancer Institute, Oregon Health & Science University, Portland, OR, USA. [5] Vaccine Research Institute, Creteil, France. [6] Department of Molecular Microbiology & Immunology, School of Medicine, Oregon Health & Science University, Portland, OR, USA. [7] Present address: Absci Corp, 1810 SE Mill Plain Blvd., Vancouver, WA 98683, USA. [8] Present address: Institute for Integrative Systems Biology, University of Valencia-CSIC, Calle Catedràtic Agustín Escardino Benlloch 9, 46980 Paterna, Valencia, Spain. ✉email: hessell@ohsu.edu

To contain the HIV epidemic, the development of longer acting antivirals with infrequent dosing requirements is of critical importance. Among other approaches[1], HIV-1 broadly neutralizing antibodies (bNAbs) isolated from B cells in elite neutralizers are being developed for clinical use[2,3]. Compared to the small molecule inhibitors comprising current anti-retroviral therapy regimens, bNAbs offer the advantages of extended half-life, reduced toxicity, and synergistic recruitment of endogenous immune components through Fc-mediated signaling.

While Fab-mediated neutralization is the dominant antiviral function of Abs to HIV-1, numerous studies demonstrate an Fc-mediated contribution, reviewed in[4]. In vitro work with human plasma provides evidence for Ab-mediated C' lysis, inactivation of virions and clearance of HIV-1[5,6] despite mechanisms developed by the virus to evade C' lysis[7]. Passive transfer of bNAbs with abolished Fc functions has provided strong evidence that effector functions, including antibody-dependent cellular phagocytosis (ADCP), cytotoxicity (ADCC), and possibly complement activity, aid protection and restrict viral proliferation. This observation has been demonstrated for bNAbs targeting various sites on the viral enveloped glycoprotein (Env), including the CD4 binding site (CD4bs), the V1/V2 glycan, and the V3 glycan[8–12].

Early mechanistic passive transfer studies in non-human primates (NHPs) with bNAb b12 showed inserting L228A and L229A (LALA), point mutations that largely abrogate binding to FcγRs and complement, reduced protection by 33% following single high-dose challenge and required fewer repeated low-dose challenges compared to unmodified b12 before breakthrough infection[8,9]. More recent studies in infected mice and NHPs found Fc functions of unmodified bNAbs contribute 21–39% to the rate of the plasma viral decay following treatment, with the remaining portion attributed to neutralization[11,12]. A noteworthy exception is bNAb PGT121, for which a pair of studies have shown loss of effector function does not impact protection[13] or influence viral decay rates in SHIV-infected macaques with or without NK cell depletion[14], despite the fact that PGT121 mediates high levels of effector function in vitro[15] and these activities may contribute to the efficacy in humanized mice[10]. Efforts to augment the contribution of effector functions through increasing Fc affinity for FcγRs have slightly improved viral outcomes in mice[12,16]. In NHPs, despite improving in vitro ADCC activity, infusion of antibody variants with enhanced FcγRIII binding has resulted in either no benefit[17] or induced NK cell necroptosis and paradoxically yielded an Fc functional null phenotype[11]. Hence, the potential of enhancing effector functions for protection or treatment remains unclear.

In this work, we show that enhancing C' activities in vitro adds no value toward reducing viremia in either blood or tissue in SHIV$_{SF162P3}$-challenged macaques pretreated with membrane external proximal region (MPER)-targeting bNAb 10E8v4[18,19]. Little is known about the in vivo effector contributions of bNAbs targeting the MPER of HIV Env. Thus, we investigated viremic outcomes with bNAb 10E8v4 containing either unmodified Fc, an Fc functional knockdown, or an Fc dually enhanced for C' functions and FcγRII/FcγRIII binding[20,21]. At low doses, treating with unmodified 10E8v4 but not the C'/FcγR dual-enhanced variant or the Fc knockdown reduces post-acute plasma and tissue virus compared to the control group. The dually enhanced variant bNAb is rapidly cleared from plasma after treatment and comparable reductions in post-acute viremia are only observed at higher doses. Analysis of the influence of effector functions measured in this study reveal an inverse correlation between 10E8v4-mediated ADCP activity in plasma at the time of challenge and the reduction in post-acute viremia among all treatment groups, but no correlations are found with either antibody-

dependent complement deposition (ADCD) or antibody-dependent complement-mediated lysis (ADCML). These results support an in vivo contribution of effector functions to the antiviral activity of MPER bNAb 10E8v4, which does not mediate ADCC against SHIV$_{SF162P3}$-infected target cells in vitro.

## Results

**Fc modifications alter 10E8v4 effector phenotype.** To determine suitable bNAb candidates for NHP studies using Fc functional alterations, we developed a panel of ten clinically relevant bNAbs with either unmodified Fc regions or with point mutations designed to abrogate or enhance FcγR binding and/or interaction with C'. Initially, we screened this panel for the ability to perform ADCML of HIV$_{BaL}$ virions and for ADCC activity against SHIV$_{SF162P3}$-infected target cells. Unmodified bNAb 10E8v4 showed comparatively high levels of ADCML against HIV$_{BaL}$, but as expected no ADCC activity against SHIV$_{SF162P3}$-infected cells[15] (Fig. 1a, b) and thus was selected to investigate the role of C' lysis in SHIV-challenged macaques. Fc variants LALA (L234A/L235A) and EFTAE (G236A/S267E/H268F/S324T/I332E; Fig. 1c), which have been previously reported to reduce or enhance, respectively, both FcγR binding and complement activation[20,21], were inserted into the IgG1 backbone of 10E8v4. These modifications do not alter binding or neutralization activities solely mediated by Fab, as variant binding to an MPER epitope containing peptide and neutralization of replication-competent SHIV$_{SF162P3}$ and pseudovirus matched those of unmodified 10E8v4 (Fig. 1d, e).

In contrast, the 10E8v4 EFTAE variant increased Fc-mediated C1q binding and ADCD (measuring C3d opsonization) by a factor of two, while 10E8v4 LALA did not bind C1q and showed three-fold less binding to ADCD (Fig. 1f, g). Consistently, 10E8v4 EFTAE showed greater ADCML of Raji B cells stably expressing either intermediate or high levels of HIV$_{YU2}$ Env on their surface[7] (Fig. 1h). Virion lysis varied by strain, however, as unmodified 10E8v4 showed strong lysis of HIV$_{BaL}$ (Fig. 1a) but little propensity for lysing SHIV$_{SF162P3}$ virions (Fig. 1i). Importantly, strong lysis of SHIV$_{SF162P3}$ was achieved with sera from five of six macaques supplemented with 10E8v4 EFTAE, suggesting the added potential for this variant to reduce infectious particles in vivo (Fig. 1i).

In addition to Fc-mediated complement functions, we measured the altered binding of LALA and EFTAE mutations on 10E8v4 to recombinant FcγR extracellular domains. The affinity of 10E8v4 EFTAE for allelic variants of solubilized FcγRII and FcγRIII measured by biolayer interferometry (BLI) was increased by a factor of 3–10 among both human and rhesus receptors (Fig. 2a). As expected, the 10E8v4 LALA mutations severely reduced or abrogated binding across human and rhesus FcγRII and FcγRIII allelic variants (Fig. 2a). Notably, the introduced Fc mutations did not alter affinity to either human or rhesus neonatal FcR (FcRn; Fig. 2b), and thus were not expected to impact FcRn-mediated IgG cellular recycling and its role in extending plasma antibody half-life[22,23]. Next, binding of fluorescently labeled 10E8v4 to FcγRs on PBMCs from four SHIV-naive rhesus macaques was measured by flow cytometry. Relative to binding of unmodified 10E8v4, 10E8v4 EFTAE showed similar binding to CD64 (FcγRI) and a 2–3 factor increase in binding to CD32 (FcγRII) and CD16 (FcγRIII) on PBMCs from three of four macaques, while 10E8v4 LALA binding to all cellular FcγRs was diminished to varying degrees (Fig. 2c). We then evaluated downstream FcγR-mediated effector functions and determined that ADCP activity reported using an MPER peptide-coated fluorescent bead-based assay was unaltered for 10E8v4 EFTAE relative to unmodified antibody, while the

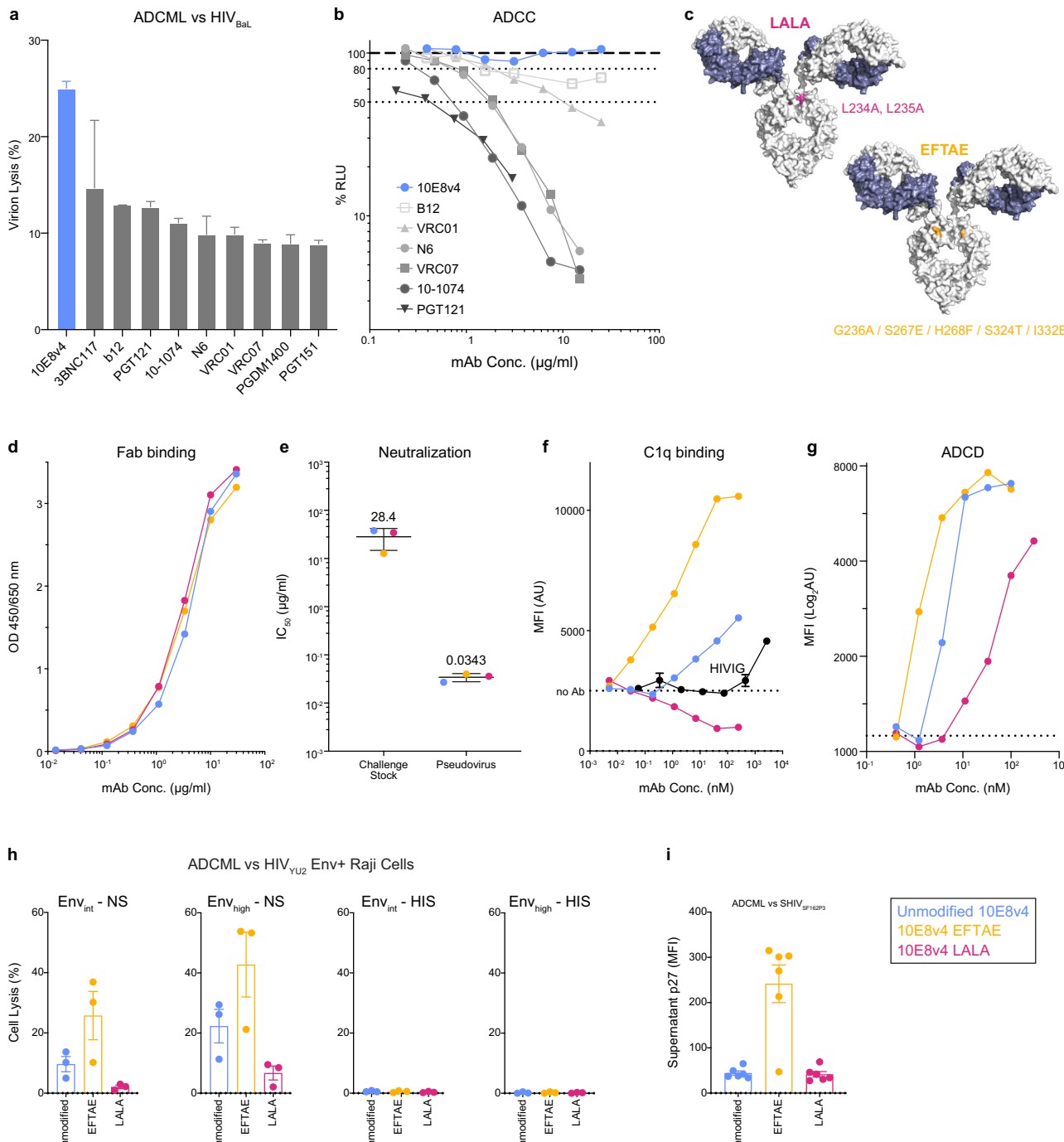

**Fig. 1 Fc variant EFTAE enhances complement-mediated functions of bNAb 10E8v4. a** C'-mediated lysis of HIV$_{BaL}$ virions incubated with a panel of bNAbs in normal serum. Data shown are mean ± SD of two replicates. Lysis was measured by flow cytometric quantification of supernatant p27 with complete lysis determined by incubation with detergent instead of antibody. **b** Comparative ADCC activity of bNAbs against SHIV$_{SF162P3}$-infected target cells expressing luciferase reporter, with greater activity corresponding to a decrease in relative light units (RLU). Data shown are mean only. **c** Site visualization of Fc point mutations inserted into each IgG1 CH$_2$ domain. Each mutation is shown on only one arm for clarity. **d** 10E8v4-unmodified and Fc-modified Fab-mediated binding to MPER determined by ELISA and **e** neutralization of SHIV$_{SF162P3}$ replication-competent challenge virus (single round infection in TZM-bl cells) and SHIV$_{SF162P3}$ pseudovirus in the TZM-bl assay. Values shown are mean ± SD among 10E8v4-unmodified and variants. **f** C1q binding and **g** ADCD assessed as C3b deposition to antibody complexed with MPER-coated beads measured by SPR. **h** Complement-mediated lysis of transduced Raji B cells resulting in either intermediate or high levels of HIV$_{YU2}$ Env surface expression. Antibody plus cells were incubated with normal (left) or heat-inactivated (right) serum and percent lysed cells determined by flow cytometry. Results are reported as the percentage of dead cells above that in wells without antibody, with biological replicates from three independent serum donors. **i** C'-mediated lysis of SHIV$_{SF162P3}$ virions incubated with 10E8v4 variants in normal serum. Data and analysis are derived from n = 6 animals per group. Analyzed data shown in **a**, **d**, **f–i** are mean ± SD and are representative of at least two independent experiments. Source data are provided in the Source Data file associated with this manuscript. The color key is shown and colors are consistent throughout the manuscript. ADCML antibody-dependent complement-mediated lysis, ADCP antibody-dependent cellular phagocytosis, ADCC antibody-dependent cellular cytotoxicity, ADCD antibody-dependent complement deposition.

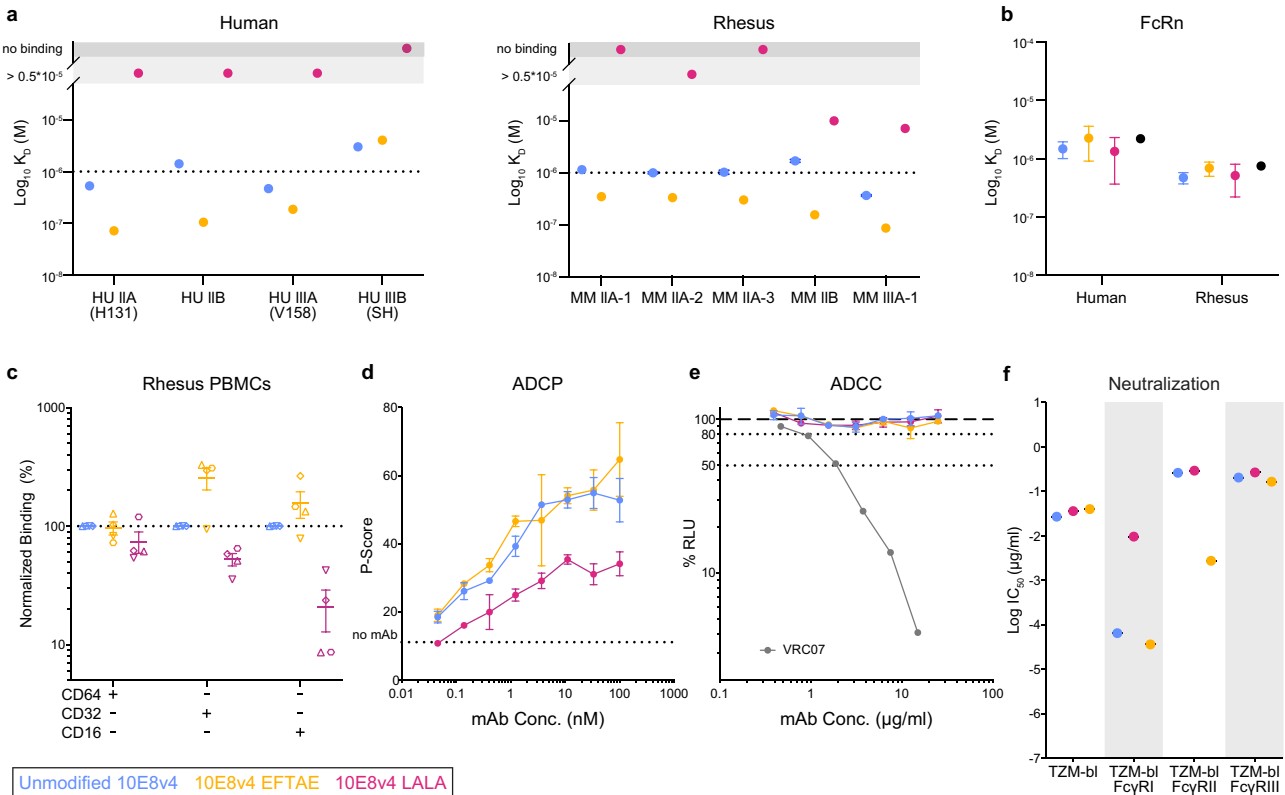

**Fig. 2 Fc variant EFTAE increases affinity for FcγRII and FcγRIII but does not alter ADCP or lack of ADCC activity. a** Kinetic affinity constants for recombinant human and rhesus macaque FcγRII and FcγRIII encoded by the indicated allelic variants determined by SPR. Dotted lines represent the highest concentration tested, above which confidence in fitted equilibrium dissociation constants is diminished. **b** Kinetic affinity constants for recombinant human and rhesus FcRn determined by SPR. Black symbols denote values of an isotype-matched mAb. **c** Fluorescently labeled antibody binding to naive PBMCs from $n = 4$ rhesus macaques determined by flow cytometry. Data are reported as the percent binding of Fc variant 10E8v4 relative to unmodified 10E8v4 for each macaque designated by matched symbols. **d** ADCP activity of 10E8v4 measured by THP-1 monocyte internalization of MPER peptide-coated fluorescent beads. **e** ADCC activity against SHIV$_{SF162P3}$-infected target cells expressing luciferase reporter, with activity measured as the loss in relative light units (RLU). CD4bs bNAb VRC07 is shown as a positive control. **f** Neutralization of SHIV$_{SF162P3}$ pseudovirus using parent TZM-bl or those transduced for surface expression of FcγRs. Analyzed data shown in **a–d** and **f** are mean ± SD and mean only in **e**. All data reported are representative of at least two independent experiments. Source data are provided in the Source Data file associated with this manuscript. The color key is shown and colors are consistent throughout the manuscript. FcRn neonatal Fc receptor, PBMC peripheral blood mononuclear cell, ADCC antibody-dependent cellular cytotoxicity.

LALA mutations resulted in a >2-fold decrease (Fig. 2d). Neither variant of 10E8v4-mediated ADCC when tested against the SHIV$_{SF162P3}$ challenge virus (Fig. 2e).

Neutralization of free virus can be expected to play a major role in antibody protection. It has been reported that FcγRI on the surface of HIV-1-susceptible cells promotes stronger neutralization by MPER specific bNAbs, likely through co-localizing these bNAbs with post attachment or budding virions displaying Env conformations favoring MPER recognition[24,25]. Here, the IC$_{50}$ of both unmodified 10E8v4 and EFTAE was 3 log$_{10}$ lower against SHIV$_{SF162P3}$ pseudovirus in FcγRI-expressing versus non-expressing target cells (Fig. 2f). Consistent with its greater affinity for FcγRII, the EFTAE variant displayed a 2 log$_{10}$ reduction in IC$_{50}$ in TZM-bl cells expressing that receptor (Fig. 2f), which suggests a potential neutralization benefit of EFTAE in addition to enhanced C'-mediated virion lysis compared to unmodified 10E8v4 vs SHIV$_{SF162P3}$.

**Pharmacokinetics of unmodified 10E8v4 and Fc variants vary in naive macaques.** We evaluated the in vivo kinetics of each of the antibodies proposed for the passive protection studies. For each antibody, we dosed three macaques by subcutaneous delivery with 10 mg/kg of either unmodified 10E8v4, 10E8v4 LALA, or

10E8v4 EFTAE. Antibody concentrations of unmodified 10E8v4 and 10E8v4 LALA peaked in plasma at day 3 or 4 in a range of 98–118 µg/ml with a half-life of 5.7 and 4.2 days, respectively. In contrast, 10E8v4 EFTAE peaked at day 1 following injection, with a plasma concentration of only 21–50 µg/ml and a half-life of 2.3 days. Re-testing 10E8v4 EFTAE in three different macaques yielded almost identical results (Supplementary Fig. 1). As most clinical trials of human bNAbs have been conducted with intravenous infusion[26], we further tested two more macaques in each 10E8v4 group to investigate if route of delivery would impact the kinetics of antibody decay in plasma. As could be expected, peak bNAb levels were slightly higher after intravenous delivery; however, the shorter half-life for 10E8v4 EFTAE was again observed (Supplementary Fig. 1).

**Single high-dose SHIV challenge study design is used for rhesus macaque passive studies.** The in vitro C' profile of 10E8v4 EFTAE, including increased C1q/C3b opsonization and enhanced virion lysis of SHIV$_{SF162P3}$ (Fig. 1f, g, i) combined with the lack of ADCC activity (Fig. 2e), made this bNAb an attractive candidate to examine the potential to improve the contributions of Fc-mediated C' activity in vivo. Since effector functions become less impactful at high neutralization titers[14,27], we selected low-dose

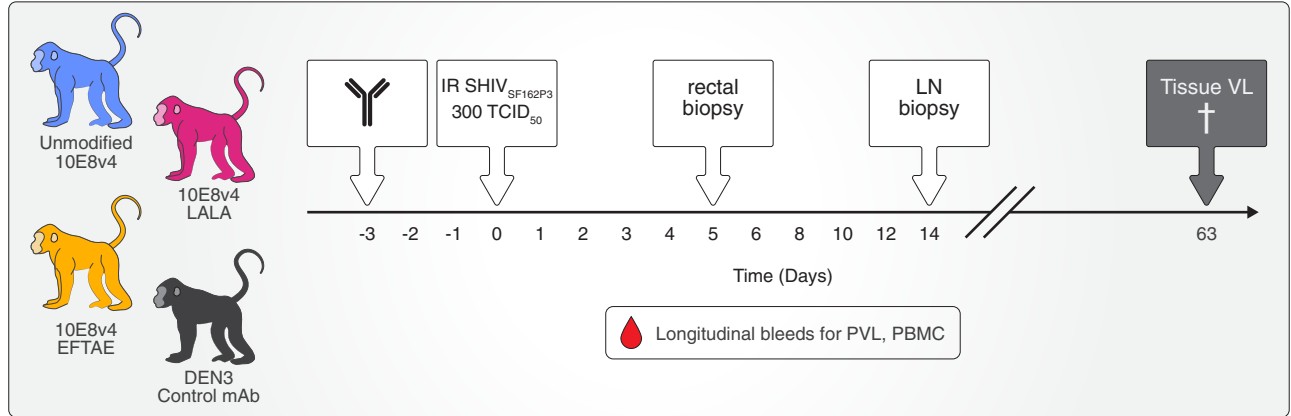

**Fig. 3 Schematic of macaque challenge model and tissue sampling.** Macaques for challenge studies were divided into four antibody treatment groups, 10E8v4 unmodified, 10E8v4 LALA, 10E8v4 EFTAE, or isotype control mAb (DEN3). Antibody was subcutaneously delivered 3 days prior to a single high-dose mucosal SHIV$_{SF162P3}$ challenge. Rectal and inguinal lymph node biopsies were collected on study day 5 and 14, respectively, PBMCs were collected every 14 days, and plasma collected weekly. Thirteen tissues representing diverse anatomical regions, detailed in Fig. 4m, were harvested on study day 63. Group colors are consistent throughout the manuscript. IR intrarectal, TCID$_{50}$ 50% tissue culture infectious dose, LN lymph node, PVL plasma viral load, PBMC peripheral blood mononuclear cells.

administration of 10E8v4 prior to challenge as our approach for the macaque protection studies. We hypothesized that in a pro-phylactic setting with incomplete neutralization, the stronger C'-mediated effector functions associated with 10E8v4 EFTAE would protect and/or control viremia superior to unmodified bNAb 10E8v4, which would, in turn, be superior to 10E8v4 LALA. Thus, we selected sub-neutralizing doses of 10E8v4 derived from a previous study using SHIV$_{BaLP4}$ that found that 10E8 plasma concentrations that were a factor of 2–3 above the in vitro IC$_{50}$ in TZM-bls were sufficient to protect[28]. Consequently, Macaques were subcutaneously treated with 5 mg/kg 10E8v4, 10E8v4 LALA, 10E8v4 EFTAE, or an isotype control IgG1 3 days prior to a single 300 TCID$_{50}$ intrarectal challenge that was expected to infect all controls (Fig. 3).

**Low-dose unmodified 10E8v4 reduces post-acute viremia in macaques.** Plasma antibody levels at time of challenge (TOC: day 0) ranged from 10 to 21, 9 to 56, and 13 to 21 µg/ml for 10E8v4, 10E8v4 LALA, and 10E8v4 EFTAE, respectively (Fig. 4a, b). There were comparable levels of functional antibody between groups at TOC as measured by neutralization titers against HIV$_{SF162P3}$ pseu-dovirus (Fig. 4c). Importantly, none of the 10E8v4-infused macaque plasma showed neutralization activity against the replication-competent virus at the time of challenge, consistent with its rela-tively high IC$_{50}$ of 28.4 µg/ml measured in TZM-bl cells (Fig. 1e). Plasma antibody half-lives were similar to those of SHIV-naive macaques, with unmodified 10E8v4 and 10E8v4 LALA showing equivalent half-lives of 5.3 days and 10E8v4 EFTAE clearing more rapidly with a 2.2-day half-life (Fig. 4a).

All macaques were productively infected by 7 days post challenge. We monitored plasma viral load (PVL) for 63 days post challenge and plotted the quantitation of SIVgag viral RNA as a function of time for a longitudinal comparison between groups (Fig. 4d, e). All groups had comparable levels of virus in plasma and inguinal lymph nodes during acute infection, with no significant difference in peak PVL (Fig. 4e, f), peak PBMC cell-associated viral load (CAVL; Fig. 4g), or inguinal lymph node CAVL at day 14 (Fig. 4h). Average post-acute PVL, however, was 86% lower (0.14 fold) in unmodified 10E8v4-treated animals compared to that in the control group ($p = 0.0012$; Fig. 4i, n). Significant differences in post-acute PVL from the control group were not found in either the Fc knockdown (LALA) or Fc enhanced (EFTAE) groups (Fig. 4i, n).

Macaques were necropsied at nine weeks post challenge and representative lymph node and gut tissues were collected, along with subsections of the spleen, liver, and reproductive tract. Among all tissues sampled, the unmodified 10E8v4 group had significantly less virus quantified as SIVgag DNA compared to controls ($p = 0.0016$; Fig. 4j), and this effect was most pronounced in the lymph nodes and spleen, where a 67% reduction (0.33 fold) compared to controls was observed (Fig. 4k, n). Tissue virus in the 10E8v4 LALA-treated group mirrored that in the control group (Fig. 4j–n), further indicating that in the absence of protective neutralizing titers at TOC, Fc functions of unmodified 10E8v4 were responsible for the reduction in post-acute viremia. Remarkably, the 10E8v4 EFTAE group had elevated levels of virus in tissues ($p = 0.035$; Fig. 4j). This was most pronounced in lymphoid tissues, where average SIVgag DNA was 103% higher (2.03 fold) compared to that of controls ($p = 0.0433$) (Fig. 4k, n), suggesting higher levels of initial reservoir seeding in those sites.

**Higher doses of FcγR/complement-enhanced 10E8v4 reduce post-acute plasma viremia.** Due to the rapid plasma decay of the 5 mg/kg dose, it was unclear if 10E8v4 EFTAE would have mediated a reduction in post-acute viremia if present at similar levels and with comparable half-life to unmodified 10E8v4 post challenge. To allow a more direct comparison with the 5 mg/kg unmodified group, as well as to investigate whether the reduced post-acute viremia observed in the unmodified group could be improved, we chose to dose additional NHPs with higher levels of 10 mg/kg (dashed lines, $n = 2$, Fig. 5) and 20 mg/kg (solid lines, $n = 2$, Fig. 5) followed by an equivalent SHIV$_{SF162P3}$ challenge. Here, plasma antibody at TOC ranged from 38 to 52 µg/ml for all four 10E8v4 EFTAE-treated animals (Fig. 5a, b) compared to 13 to 21 µg/ml in those treated with 5 mg/kg (Fig. 4b). Unexpectedly, with higher dosing, there was slightly more disparity in plasma antibody between groups at TOC, with 10E8v4 EFTAE con-centrations and neutralization titers two to three times lower than those in the unmodified group (Fig. 5b, c).

Longitudinal plasma viremia was reduced in all high-dose groups compared to that in the control group, with 20 mg/kg dosing resulting in greater suppression of post-acute viremia than in the 10 mg/kg groups (Fig. 5e). Peak plasma viremia was lower overall in both treatment groups, although one macaque (30192) treated with 20 mg/kg 10E8v4 EFTAE experienced peak viremia

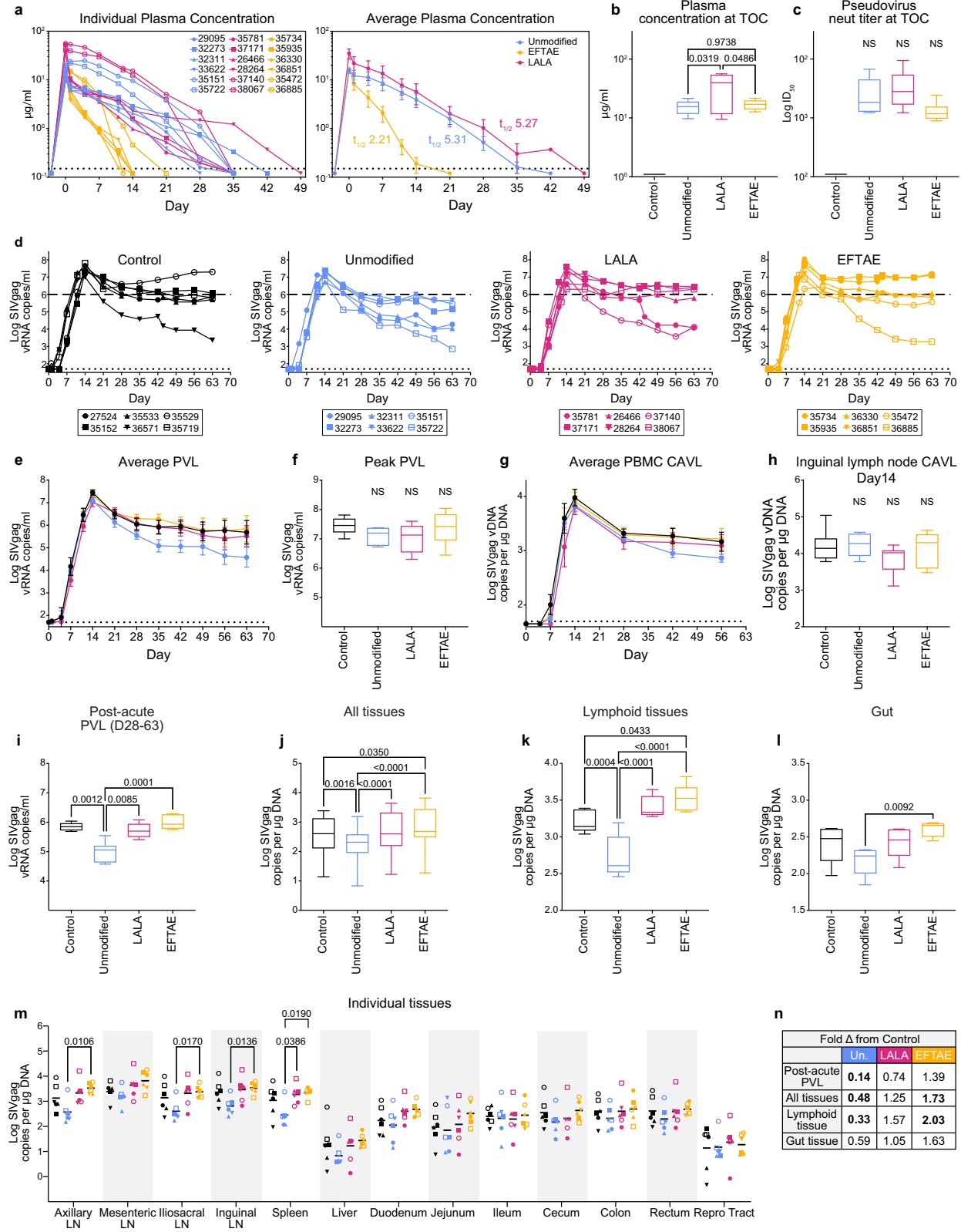

comparable to controls (Fig. 5e, f). Average post-acute PVL in both the unmodified and EFTAE groups was lower by 88% (0.12 fold; $p < 0.0001$) and 78% (0.22 fold; $p = 0.0072$), respectively, than that of control group (Fig. 5h). Taken together with the results from the 5 mg/kg treatment groups (Fig. 4), these results show that sub-protective levels of unmodified 10E8v4 present at the time of challenge reduced post-acute viremia by 78–88% and this effect was dependent on intact Fc functions (Supplementary Fig. 3). However, dually enhancing FcγR binding and C' did not improve this outcome and, on the contrary, may have encouraged tissue seeding when the antibody was present at low levels (<20 μg/ml in plasma; Fig. 4j–n).

**Fig. 4 Pre-treatment with 5 mg/kg unmodified but not FcγR/complement-enhanced 10E8v4 decreases post-acute plasma and tissue virus in macaques. a** Longitudinal or **b** TOC plasma bNAb concentrations following 5 mg/kg 10E8v4 subcutaneously delivered 3 days prior to challenge (day −3). The average half-life ($t_{1/2}$) ± se calculated from the $n = 6$ animals in each group is shown in the right panel of Supplementary Fig. 2a, c Plasma neutralization titers against SHIV$_{SF162P3}$ pseudovirus at TOC. **d** Individual and **e** average ± se longitudinal plasma viremia of macaques in each bNAb treatment group. **f** Average peak viremia, **g** longitudinal PBMC CAVL ± se, and **h** day 14 inguinal lymph node CAVL. **i** Average post-acute PVL defined as days 28–63 post challenge. **j** Average tissue viral load at necropsy (day 63) from all tissues, **k** lymphoid tissues (lymph nodes and spleen), or **l** gut tissues. **m** Individual tissue viral load at necropsy. **n** Summary of fold-difference in viral load from the control group, with statistically significant comparisons ($p < 0.05$) in bold. Boxes in box and whisker plots extend from 25 to 75 percentiles with a line at median and whiskers extending to min–max values. Statistical comparisons were performed using a one-way ANOVA (**b**, **c**) followed by Tukey's post-hoc comparison between groups and Dunnett's post-hoc test for comparison to control group (**f**, **h**), two-way ANOVA (**i**, **j**, **k**, **l**), and two-way repeated-measures ANOVA (**m**) followed by Tukey's post-hoc comparison between groups. Viral loads are presented (**d**, **e**) and analyzed using log$_{10}$ transformed data. All <0.05 adjusted $p$ values are shown as well as non-significant $p$ values between comparisons mentioned in the text. NS designates no significant difference from any group. Data and analysis are derived from $n = 6$ animals per group with symbols denoting individual macaques as indicated in (**d**) and also used in (**m**). Group colors are consistent throughout the manuscript. Statistical significance was determined at the significant alpha level of 0.05 and performed in GraphPad Prism 9. Data shown in panels **a**, **b**, **c**, **e**–**m** are representative of at least two independent experiments. Source data are provided in the Source Data file associated with this manuscript. TOC time of challenge, PVL plasma viral load, PBMC peripheral blood mononuclear cell, CAVL cell-associated viral load.

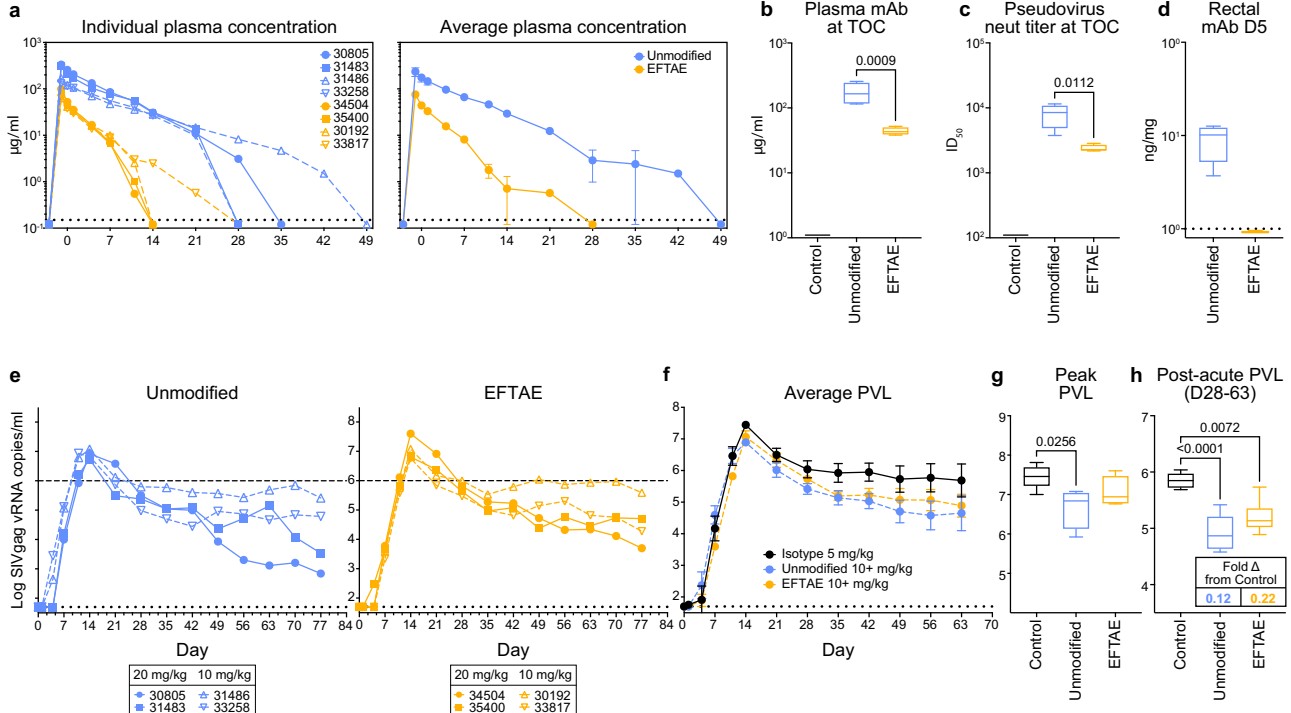

**Fig. 5 Pre-treatment with 10 and 20 mg/kg unmodified 10E8v4 and 10E8v4 EFTAE decreases post-acute plasma viremia in macaques. a** Longitudinal or **b** TOC plasma bNAb concentrations following 10 (dashed lines) or 20 (solid lines) mg/kg 10E8v4 subcutaneously delivered 3 days prior to challenge (day −3). The average half-life ($t_{1/2}$) ± se calculated from the $n = 4$ animals in each group (Supplementary Fig. 2) is shown in the right panel of (**a**). **c** Plasma neutralization titers against SHIV$_{SF162P3}$ pseudovirus at TOC. **d** bNAb concentrations in biopsies collected from the rectal mucosa 5 days after challenge (8 days post delivery). **e** Individual and **f** average ± se longitudinal plasma viremia of macaques in each mAb treatment group. A dashed line at 10$^6$ vRNA copies/ml is presented in (**e**) as a visual reference. Average **g** peak or **h** post-acute PVL, defined as from day 28–63 with a table insert showing fold-difference compared to the control group. Boxes in box and whisker plots extend from 25 to 75 percentiles with a line at median and whiskers extending to min–max values. Statistical comparisons were performed using a one-way (**b**, **c**, **d**, **g**) or two-way ANOVA (**h**) followed by Tukey's post-hoc test between groups. All <0.05 adjusted $p$ values are shown. Boxes in box and whisker plots extend from 25 to 75 percentiles with a line at median and whiskers extending to min–max values. Statistical comparisons were performed using a one-way ANOVA (**b**, **c**) followed by Tukey's post-hoc comparison between groups and Dunnett's post-hoc test for comparison to control group (**g**), two-way repeated-measures ANOVA (**h**) followed by Tukey's post-hoc comparison. Data and analysis are derived from $n = 4$ animals per group with symbols denoting individual macaques as indicated in (**e**). Viral loads are presented and analyzed using log$_{10}$ transformed data. Group colors are consistent throughout the manuscript. Statistical significance was determined at the significant alpha level of 0.05 and performed in GraphPad Prism 9. Data in **b**–**d**, **g**, and **h** are representative of at least two independent experiments. Source data are provided in the Source Data file associated with this manuscript. TOC time of challenge, PVL plasma viral load.

**Plasma ADCP activity at TOC inversely correlates with the reduction in post-acute viremia**. To investigate possible correlates with the reduction in post-acute viral load, we measured plasma activity of passively transferred bNAb at TOC as well as

longitudinal metrics of endogenous immune responses. As noted above, neutralizing activity from passively administered bNAb was not detected in the plasma of any animal at TOC against the challenge virus. However, plasma ADCP activity at TOC,

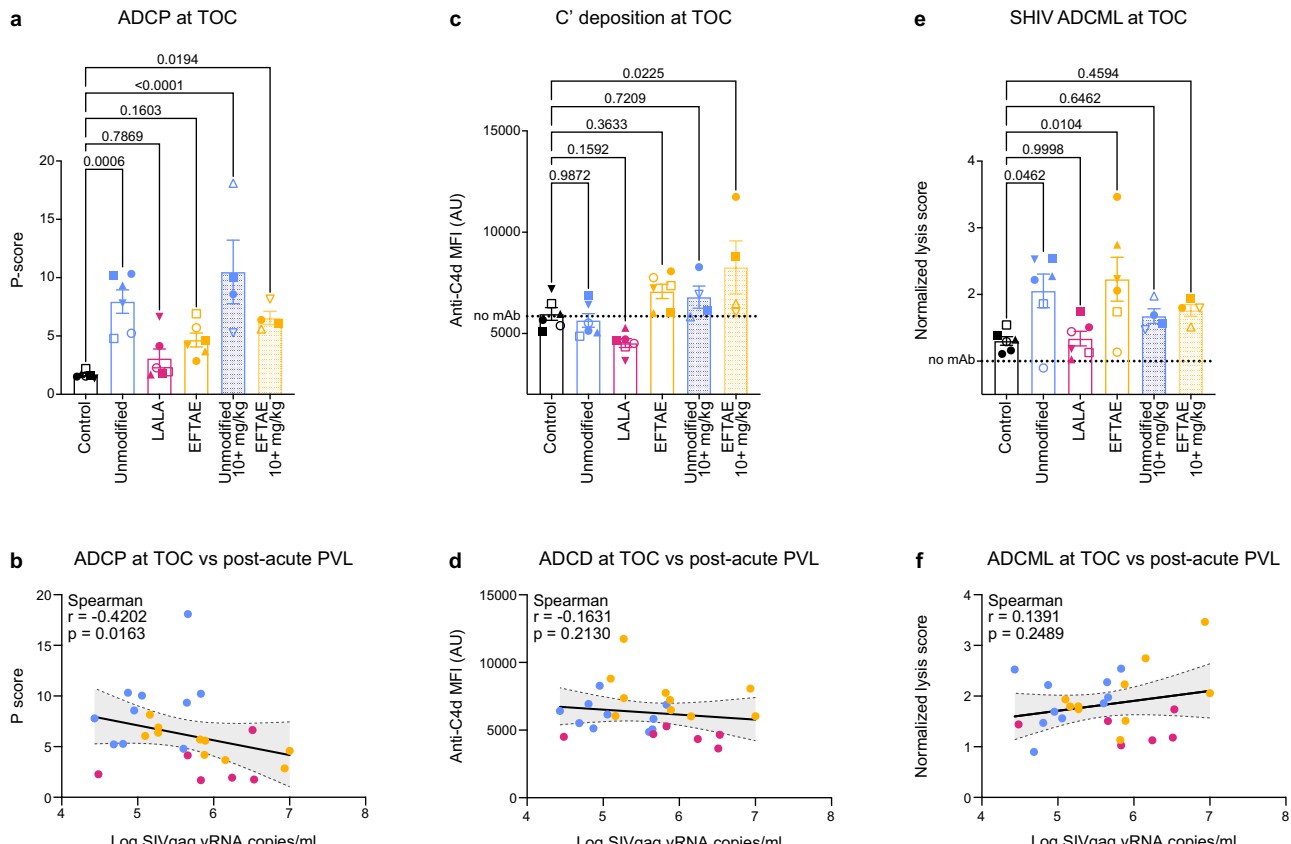

**Fig. 6 Effector activity in plasma at TOC. a** Plasma ADCP activity determined by co-incubation with THP-1 monocytes and measured as internalization of MPER-coated fluorescent beads. **c** C3b deposition to plasma antibody complexed with MPER-coated beads measured by SPR. **e** C'-mediated lysis of SHIV$_{SF162P3}$ virions incubated with heat-inactivated plasma supplemented with normal serum. Error bars in **a**, **c**, and **e** represent average values ± se with $p$ values from one-way ANOVA followed by Dunnett's post-hoc test. **b**, **d**, **f** Correlation matrices between the indicated effector function and post-acute PVL for each individual 10E8v4-treated macaque, with $r$ and $p$ calculated from one-tailed Spearman tests. Error band denotes best fit values ± se with 95% confidence intervals. Symbols for animal IDs in **a**, **c**, and **e** are the same as those shown in Figs. 4 and 5. Data and analysis are derived from $n = 6$ animals per group (5 mg/kg); $n = 4$ animals per group (10+ mg/kg). Statistical significance was determined at the significant alpha level of 0.05 and performed in GraphPad Prism 9. Data are representative of at least two independent experiments. Source data are provided in the Source Data file associated with this manuscript. Animal symbols and group colors are consistent throughout the manuscript. ADCP antibody-dependent cellular phagocytosis, C' complement, ADCML antibody-dependent complement-mediated lysis, TOC time of challenge, PVL plasma viral load.

measured in vitro by THP-1 internalization of MPER peptide-coated beads, was significantly higher than that of controls among groups with reduced viremia (Fig. 6a), and there was an inverse correlation between ADCP at TOC and post-acute viral load (Spearman $r = -0.4202$, $p = 0.0163$; Fig. 6a, b). There was no correlation between complement deposition and ADCML (Fig. 6c–f).

**bNAb delivery does not substantially alter endogenous immune responses**. It has been reported that bNAb delivery in the context of HIV/SHIV may augment the development of native immune responses through immune complexes driving activation of antigen-presenting cells[29]. In this study, passive transfer of unmodified 10E8v4 did not significantly alter the relative composition of effector cells (Fig. 7a and Supplementary Fig. 4). Autologous binding titers to SF162 gp140 among macaques that seroconverted were generally similar within and across treatment groups (Fig. 7b, c), although it should be noted that 5 of the 32 macaques did not seroconvert. Two of these were in the control group and one was from the 5 mg/kg EFTAE group where seroconversion noticeably varied in titer and time of onset (Fig. 7c). Overall, neither serostatus nor binding titers were associated with a difference in post-acute PVL. In addition, no

endogenous neutralizing activity was detected in any animal on days 28, 42, and 56 against the challenge virus (Fig. 7d). Virus-specific CD4$^+$ and CD8$^+$ T cell responses were assessed at multiple time points, and there was no clear association between specific T cells and post-acute viral load (Fig. 7e, f and Supplementary Fig. 5).

**Neither mucosal targeting nor anti-drug antibodies (ADA) account for rapid plasma clearance of EFTAE**. Given the higher affinity of 10E8v4 EFTAE for FcγRII and FcγRIII (Fig. 2a), it is possible that at least part of the lower plasma concentrations compared to unmodified or LALA variant was due to increased cellular affinity and/or altered tissue trafficking. While we were not able to directly detect 10E8v4 on the surface of PBMCs in any group 1–3 days post delivery, 10E8v4 was detected in day 5 rectal biopsies of NHPs receiving 10 or 20 mg/kg unmodified bNAb (Fig. 5d). Concentrations for unmodified 10E8v4 measured near 10 ng/mg of rectal tissue, corresponding to a plasma to rectal mucosa concentration ratio ranging from 7 to $21 \times 10^3$:1 (Supplementary Fig. 2b). In contrast, antibody in rectal mucosa was undetectable in the 10 or 20 mg/kg dose 10E8v4 EFTAE groups or in any 5 mg/kg dose groups (<1 ng/mg), as would be expected if both variants were present in plasma/rectal mucosa

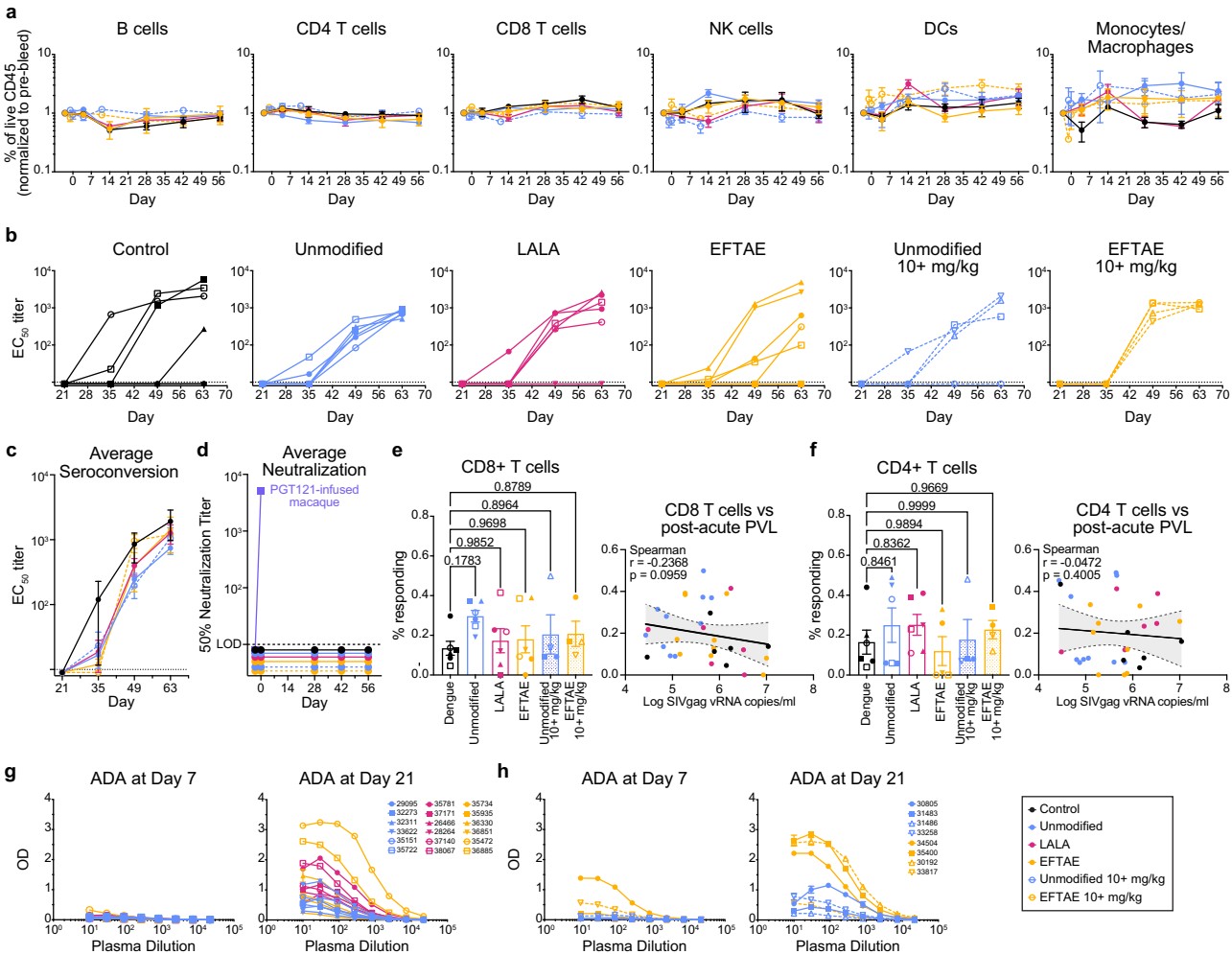

**Fig. 7 Endogenous immune responses and ADA in macaques. a** Mean ± se of cell subsets as the percent of live CD45+ in each treatment group normalized to the percentage composition prior to bNAb treatment. **b** Individual or **c** mean ± se longitudinal plasma binding titers to HIV$_{SF162}$ gp140 measured by ELISA. **d** Average longitudinal neutralization IC$_{50}$ values from each group against the replication-competent SHIV$_{SF162P3}$ challenge stock measured in the TZM-bl assay. Plasma from a macaque infused with bNAb PGT121 was used as an assay control. **e, f** Intracellular cytokine staining on PBMCs collected from study day 56 and incubated with SIV$_{mac239}$ gag pooled peptides. Data are reported as the percent CD8+ or CD4+ of the CD3+ CD45RA- IFNγ+ population in peptide-stimulated minus that in mock-stimulated cultures. Individual values are plotted by treatment group, with bars showing mean ± se, and on correlation matrices with post-acute viral load, where r and p are derived from one-tailed Spearman tests and visualized with a simple linear regression line. **g** Plasma from each macaque treated with 5 mg/kg and **h** 10 or 20 mg/kg 10E8v4 was tested for the development of ADA 10 days (study day 7) and 24 days (study day 21) after injection via ELISA. Statistical comparisons were performed using repeated-measures one-way ANOVA (**a, e, f**) with multiple comparisons followed by Dunnett's post-hoc test, and no statistically significant differences were found between treatment and control groups. Data and analysis are derived from n = 6 animals per group (5 mg/kg); n = 4 animals per group (10+ mg/kg). Statistical significance was determined at the significant alpha level of 0.05 and performed in GraphPad Prism 9. Symbols denoting individual macaques in **e–g** are shown in Figs. 4 and 5. Animal symbols and group colors are consistent throughout the manuscript. Data are representative of at least two independent experiments. Source data are provided in the Source Data file associated with this manuscript. ADA anti-drug antibodies.

concentration ratios similar to that of unmodified 10E8v4 (Fig. 5d and Supplementary Fig. 2). Thus, it is unlikely that 10E8v4 EFTAE was preferentially routing to the mucosa.

While generally well tolerated, some macaques developed ADA to human IgG1. Here, we found minimal ADA (OD < 2) occurred in NHPs receiving unmodified 10E8v4 or 10E8v4 LALA (Fig. 7g, h). Five of the ten 10E8v4 EFTAE-treated animals developed modest ADA responses by study day 21 (24 days after delivery). Only two of these had detectable ADA on day 7 (Fig. 7g, h), and thus ADA did not account for the substantially reduced 10E8v4 EFTAE levels observed by that time point (Figs. 4a and 5a).

**10E8v4 EFTAE can mediate antibody-dependent infection enhancement (ADE) in vitro in the absence of lysis**. The role of

antibody-mediated complement activity in HIV infection remains controversial, with in vitro data suggesting both pro- and antiviral contributions[7,30–32]. To further investigate the slightly enhanced levels of virus in lymphoid tissues in the 5 mg/kg 10E8v4 EFTAE group (Fig. 4j, k, n), we sought to determine its potential for exacerbating viremia due to ADE. Using a model with primary rhesus splenocytes enriched for either T cells or monocyte-derived dendritic cells (MDCs), we found no evidence of ADE in the presence of normal serum (Supplementary Fig. 6a, b). Within MDC-enriched cultures that express high levels of complement receptor-2 (CR2), however, pre-coating virus with heat-inactivated serum followed by spinoculation of cells pretreated with a low level (5 μg/ml) 10E8v4 EFTAE showed a dramatic and reproducible enhancement of cell-associated virus

(Supplementary Fig. 6a, c). Consistent with a hypothesized mechanism of surface co-localization of 10E8v4 and virus mediated by C' receptors, cultures treated with two C' knock-down variants of 10E8v4, LALA and K322A (KA), showed reduced cell-associated virus compared to unmodified 10E8v4. This effect did not occur in splenocyte cultures enriched for T cells (Supplementary Fig. 6b). Together, these data show that within cell environments with high levels of CR2 expression and in the absence of C'-mediated lysis, 10E8v4 EFTAE may facilitate ADE, although it is unclear whether such conditions occur in vivo.

The data presented here show that a single sub-protective dose of 10E8v4 reduces post-acute plasma and tissue virus in macaques challenged with SHIV$_{SF162P3}$. This reduction was dependent on intact Fc effector functions and inversely correlated with FcγR-mediated ADCP by 10E8v4 at TOC. The functional enhancement of C'-mediated virion lysis did not improve post-acute viremic outcomes, and instead, enhanced C' functions with sub-optimal levels of 10E8v4 were associated with slightly higher levels of tissue viral DNA in lymphoid tissues.

## Discussion

The first efficacy trial for bNAb given as pre-exposure prophylaxis using moderately potent bNAb VRC01 provided proof-of-concept that bNAbs can protect against susceptible viral strains[33]. The multiplicity of circulating viral strains, with high genetic diversity within (15–20%) and between (25–35%) subtypes[34], will necessitate multi-epitope targeting with cocktails of 2–3 bNAbs and/or engineered bi/tri-specific Abs with heterologous Fab domains[35,36]. In addition, maximizing effector contributions against neutralization insensitive viruses may be needed to achieve widespread efficacy. Since the initial report in 2007 on the importance of Fc receptor binding in antibody-mediated prophylaxis for HIV[8], the relative antiviral contribution of effector functions of bNAbs has garnered significant interest. Subsequent loss-of-function studies in both humanized mice and NHPs have confirmed an important role for antibody-FcR interaction, demonstrating a 21–39% contribution toward plasma viral decay slopes following treatment of established infection[4,11,12].

A critical revelation from these studies exposes the contextual dependency of effector contributions, with inconsistencies occurring between different bNAb/virus combinations and, in the case of PGT121, a negligible contribution against SHIV$_{SF162P3}$ in macaques[13,14]. This disparity is likely determined by multiple variables impacting FcγR engagement and crosslinking in vivo, including (1) steric hindrance to FcγRs, determined by bNAb epitope and angle of approach[37]; (2) number of bNAb molecules coating the virion, determined by bNAb dose, affinity for Env, and Env frequency and proximity on the viral surface[38]; and (3) affinity of FcγRs for immune complexes, which is influenced by isotype, subclass, and Fc glycosylation[39] and exhibits allele-dependent fluctuation at both a species and individual level[40,41].

Loss-of-function studies to date have used bNAbs with moderate to high levels of ADCC activity, potentially obscuring the respective contributions of ADCC, ADCP, and C' functions. Here, we report the first macaque study using Fc-modified variants of bNAb 10E8v4 targeting the MPER region of HIV-1 Env that shows no ADCC activity against the challenge virus in a cell-based SHIV infection in vitro assay using macaque CD16 trans-duced cells as effectors[15,42]. At sub-neutralizing concentrations against the challenge virus, we find that other intact effector functions of unmodified 10E8v4 are associated with a ~1 log$_{10}$ or 78–88% reduction in average post-acute viremia when neutralization is absent or insufficient to protect. In contrast, the effector null Fc variant of 10E8v4 resulted in post-acute PVL and

tissue viremia that matched control viremia. The inverse correlation between post-acute PVL and 10E8v4-mediated ADCP activity at TOC, together with the lack of such association between 10E8v4-mediated C' deposition or lysis, or between endogenous IgG or T cell responses, suggests that this effect may be the result of phagocytic clearance mediated directly through FcγRs that reduces tissue seeding during the early stages of infection[43,44]. There is evidence that neutrophils and macrophages are among the first cell types to bind HIV-1 in the early minutes to hours of transmission at mucosal surfaces[45] and macrophages and DCs subsequently become associated with HIV-1 in germinal centers as the initial CD4 T cell reservoir is established within the first few days[43]. In our study, a possible mechanistic explanation for the reduced post-acute viremia is that early monomeric recognition of 10E8v4-bound SHIV by FcγRI on phagocytes reduced initial exposure of CD4$^+$ T cell targets, and thereby lessened initial reservoir seeding and miti-gated post-acute viral loads[46].

A limitation of this study and similar pre-clinical in vivo stu-dies in animal models is that while precise effector functions can be qualitatively compared by use of in vitro assays, these assays do not comprehensively reflect in vivo conditions and may conse-quently bias interpretation toward activity favored in vitro. For example, Ab activity can vary substantially across the numerous ADCC assays that have been described[47,48]. Nevertheless, here, we can reasonably conclude that in the absence of ADCC and neutralization, the reduction in post-acute viremia was dependent on other intact effector functions across groups. As a single factor, the modest inverse correlation between phagocytosis (MPER-coated beads in vitro) at TOC and reduced viremia (SIV viral RNA in vitro) suggests the possible importance of in vivo pha-gocytic activity on subsequent viremia. While such correlative relationships cannot establish causation, it is noteworthy that ADCP from vaccine-induced gp140-specific Ab has been linked to protection or delayed SIV acquisition and lower peak viral loads in macaques[49–54]. Similarly, vaccine-induced anti-gp41 and anti-gp140 IgG-mediated antibody-dependent monocyte phago-cytosis and FcγRIIa engagement correlate with reduced HIV-1 acquisition risk in humans[55]. As human and rhesus peripheral blood monocytes display equivalent levels of ADCP with human IgG1 bNAb in vitro[56], these prior studies, combined with the data presented here underscore a prospective clinical benefit of administering bNAbs with favorable ADCP activity.

An outstanding question is whether Fc engineering can further improve effector contributions. Evidence exists for greater ADCC activity in vitro corresponding to better protection in humanized mice[12,57], but this outcome has not been recapitulated in NHPs where hyperactivation of FcγRIII$^+$ cells resulted in necroptosis and a net loss of ADCC[11]. Here, we sought to improve antiviral Fc contribution in the 10E8v4 EFTAE group, where the point mutations yield an in vitro phenotype with three potential advantages: (1) the addition of C'-mediated virion lysis; (2) faster endocytosis of immune complexes through FcγRIIb[58]; and (3) increased neutralization potency at the surface of FcγR$^+$ cells. Of these, we deemed the addition of lytic activity the most likely to affect viremia in our model, as FcγRIIb-mediated clearance in mice[58] has yet to be shown in NHPs and 10E8v4 weakly neu-tralizes SHIV$_{SF162P3}$. It is important to note that while Hessell et al. previously concluded that loss of C1q binding does not alter protection with bNAb b12[8], more recent work nuances this conclusion by demonstrating b12 does not mediate end-stage lysis of either virions or infected cells[59]. We found that the functional addition of ADCML in vitro with 10E8v4 EFTAE did not improve in vivo viremic outcomes over that in the 10E8v4-unmodified group. On the contrary, while higher dosing resulted in a clear but smaller reduction in post-acute viremia (78%) than

that found with high doses of unmodified 10E8v4, the 5 mg/kg 10E8v4 EFTAE group showed markedly elevated (103%) end-point viral loads in lymphoid tissues compared to that in controls, and, in the absence of lysis, was associated with ADE under high CR2-expressing culture conditions in vitro. Whether 10E8v4 EFTAE-opsonized, CR-mediated ADE occurred in vivo is unclear, but a similar mechanism has been described in other settings[30]. The pro- and antiviral roles of complement in HIV infection are complex and still being elucidated[7,31,60]; however, we conclude from these data that antibody-mediated C' enhancement is unlikely to improve bNAb efficacy and that antibody-mediated C'-knockout studies to mitigate possible ADE are warranted.

In conclusion, the data presented here show bNAb-mediated ADCP activity associated with reduced post-acute viremia in a neutralization insensitive context. Furthermore, they provide a rationale for investigating ADCP effector enhancements, particularly by increasing accessibility to FcγRs by utilizing IgG1 hinge extensions or the IgG3 subclass[61] to circumvent unintended consequences of point mutations enhancing FcγR affinity. Together, these efforts advance bNAb cocktail design toward achieving efficacious prophylaxis against the global diversity of HIV-1 strains.

## Methods

**Animals and virus**. Rhesus macaques of Indian origin were co-housed at the Oregon National Primate Research Center and all care and experimental protocols were approved by the Institutional Animal Care and Use Committee at Oregon Health and Science University. Macaques double negative for MHC alleles *Mamu-B*08* and *Mamu-B*17* were selected for challenge experiments. The ages of animals ranged from 2 to 10 years. Both male and female animals were used in this study. Groups were balanced for gender and age. Viral challenges were performed by intrarectal delivery of 1 ml diluted virus (ARP-6526, Lot# 170117) corresponding to 300 $TCID_{50}$ in rhesus PBMCs.

**C1q binding and C3 deposition (ADCD)**. Analytical flow cytometry was used to assess the ability of the 10E8v4 panel to recruit C1q and activate complement on the surface of antigen-coated beads. Antigen-coated beads were prepared by covalently coupling NeutrAvidin protein (31000, ThermoFisher Scientific) to coded MagPlex® superparamagnetic carboxylated magnetic microparticles (Luminex Corp.) using carbodiimide crosslinking chemistry as previously described[62], and then mixed with a 34-residue biotinylated MPER peptide (RRR-NEQELLELDK-WASLWNWFDITNWLWYIR-RRR-biotin, GenScript) containing the linear epitope recognized by 10E8v4[18] at 100 nM for at least 20 min at room temperature.

Assessment of C1q binding was carried out as previously described[62]. Briefly, antibodies were first incubated with beads overnight at 4 °C with shaking. Plates were washed five times on an automated magnetic plate washer (405, BioTek) with assay buffer (1X PBS, 0.1% BSA, 0.05% Tween-20) and subsequently incubated with 1.0 μg/ml biotinylated human C1q for 1 h at RT, followed by washing and incubation with 1:500 streptavidin-PE (PJRS25, Prozyme) for 30 min at RT. Beads were washed and resuspended in xMAP Sheath Fluid (Luminex Corp) before acquisition on the MAGPIX® System (Luminex Corp.).

For C3d deposition (ADCD), human complement serum (S174, Sigma-Aldrich) diluted 1:100 in gelatin veronal buffer supplemented with $Ca^{2+}$ and $Mg^{2+}$ ($GVB^+$, G6514, Sigma-Aldrich) was incubated with antibody-complexed antigen beads for 30 min at 37 °C, placed on ice to stop complement activation reactions, washed, and incubated with 0.1 μg/ml biotinylated anti-human C3d (A702, Quidel) for 1 h at RT with shaking. Following washes, bound anti-C3d antibodies were detected by incubation with 1.0 μg/ml streptavidin-PE (PJRS25, Prozyme) for 20 min at RT, washed, and data reported as median fluorescent intensity (MFI) were acquired on the FLEXMAP 3D® system (Luminex Corp.).

Data shown are representative of at least two independent experiments. Assay wells containing assay buffer in lieu of antibody were used to assess background C1q association or complement deposition driven by antibody-independent pathways.

**Virion ADCML**. Complement lysis of $SHIV_{SF162P3}$ virions was assessed by measurement of capsid protein p27, released following viral membrane disruption as previously described[63]. In 96-well polystyrene tissue-culture-treated microplates (6916A05, Corning), 2 ng/ml [p27] of $SHIV_{SF162P3}$ virus and a 1:50 dilution of human complement serum (S1764, Sigma-Aldrich) were mixed with IgG in $GVB^{++}$ for a total volume of 150 μl. To generate a p27 standard curve for interpolation of percent lysis, disruption buffer (5421, ABL, Inc.) was added at a 1:10 dilution to serially-diluted virions, whereas other samples received an equivalent volume of

$GVB^{++}$. Heat-inactivated complement serum (56 °C, 30 min) and wells containing active complement serum without antibody served as negative controls for baseline p27 concentrations and complement-mediated lysis via antibody-independent pathways, respectively. Plates were incubated at 37 °C for 1.5 h with gentle shaking before transferring 80 μl to black 96-well clear flat-bottom plates (655906, Greiner Bio-One) for quantification.

Quantification of released p27 was carried out using a bead-based sandwich assay. Briefly, MagPlex® beads covalently coupled to an anti-SIVmac251 p27 monoclonal antibody (ARP-13443, HIV Reagent Program) were incubated with each sample for 1 h at RT with gentle orbital shaking (600 rpm), followed by five washes on an automated plate washer. The degree of p27 bound to the beads was detected via incubation with 1.0 μg/ml biotin-anti-SIVmac p27 (ARP-1610, HIV Reagent Program) for 1 h at room temperature with shaking, and subsequent staining with streptavidin-PE (PJRS25, Prozyme). After incubation and washing steps, beads were resuspended in xMAP® sheath fluid (Luminex Corp), and MFI values were recorded by the MAGPIX® System (Luminex Corp).

For bNAb prospecting, 2 ng/ml [p24] of aldrithiol-2-inactivated HIV-1_{BaL} virus was used, and released p24 was captured using MagPlex® beads conjugated with two monoclonal murine anti-p24 antibodies (ab9072 and ab9044, Abcam), detected via 0.5 μg/ml polyclonal rabbit anti-p24 (NBP2-41214, Novus Biologicals), and stained with 0.6 μg/ml R-PE-conjugated rat-anti-rabbit Ig (4065-09, Southern Biotech). Data reported as mean and standard deviation of three technical replicates, and are representative of at least two independent experiments.

**ADCML of Raji cells**. Lysis of Raji cells transduced to express HIV-1_{YU-2b} Env was assessed as described previously[7]. Transduced Raji cells were sorted for Env expression via a GFP reporter signal followed by clonal expansion to obtain cells expressing intermediate or high Env levels. Env⁺ cells were then mixed with 50% normal human serum or 50% heat-inactivated human serum and 15 μg/ml 10E8v4 for 24 h at 37 °C. Complement-mediated lysis was measured with live/dead fixable aqua dead cell marker (L34957; Life Technologies) prior to fixation and flow cytometric analysis (Attune NxT; Invitrogen). Results are reported as the percentage of dead cells above that in wells without antibody, with biological replicates from three independent serum donors.

**Affinity to soluble FcγRs and FcRn**. BLI using the Octet RED96 system (Sartorius AG) was used to characterize the 1:1 biophysical interaction between 10E8v4 antibodies with human Fc gamma receptors (FcγRIIA-H131, FcγRIIB, FcγRIIIA-V158, and FcγRIIIB-SH[40]), rhesus Fc gamma receptors (FcγRIIA-1, FcγRIIA-2, FcγRIIA-3, FcγRIIB, FcγRIIIA-1[41]), and both human and rhesus neonatal Fc receptors (FcRn). For the measurement of biophysical interaction between the test antibodies with FcRn, protocols provided by the manufacturer of biotinylated recombinant human (AcroBiosystems, Cat. #FCM-H82W4) and rhesus (Acro-Biosystems, Cat. #FCM-C82W5) FcRn were followed.

All kinetic experiments were performed in freshly prepared and filtered kinetics buffer (1X PBS, 0.1% BSA, 0.05% Tween-20) at 30 °C. Biotinylated receptors were immobilized using streptavidin-coated biosensors (Sartorius AG, Item #18-5019) via a loading step with a 0.3 nm response unit threshold. Following a 60-s baseline step in kinetics buffer, loaded biosensors were dipped into two-fold serially-diluted antibody samples (1000–15.63 nM) for a 60-s association step and subsequently for 60 s in kinetics buffer to measure dissociation. Immobilized receptors were regenerated between antibody analytes by dipping 3 × 5 s into regeneration buffer (10 mM Glycine, pH 1.7).

Binding sensorgrams were aligned to the beginning of the association step for inter-step correction and Y-aligned to the pre-association baseline step, and following single reference subtraction consisting of immobilized receptor dipped into kinetics buffer, processed sensorgrams were globally fit to a 1:1 binding isotherm on ForteBio HT Analysis Software (version 11.1.1.39) to determine kinetic constants. Two experiments using streptavidin-coated biosensors (Sartorius AG, Item #18-5019) were conducted to account for variability between sets of biosensors.

**10E8v4 binding to rhesus PBMCs**. To determine bNAb binding to FcγRs on PBMCs, fluorophores were directly conjugated to 10E8v4 (PacBlue), 10E8v4 LALA (APC), and 10E8v4 EFTAE (FITC) using standard antibody labeling kits (Invitrogen). Cells from four different rhesus macaques were divided into cluster tubes (Corning; 1 × 10⁵ cells/tube) and washed twice with staining buffer (1 ml phosphate-buffered saline supplemented with 1% fetal bovine serum (FBS) and 1 mM EDTA). Live/Dead dye (Fixable Blue, Invitrogen) was added according to the manufacturer's directions and incubated for 15 min at 4 °C protected from light. After washing, 10 μg/ml of each bNAb was added to separate cluster tubes containing PBMCs from each animal, along with surface markers CD64:BV605, CD32:PE, and CD16:BV711 then incubated for 30 min as before. Cells were then washed twice and analyzed on a FACSymphony A5 (BD Biosciences). Data were analyzed in FlowJo and reported as the percent of live single lymphocytes binding the bNAb variant of interest normalized to the percent of cells binding unmodified 10E8v4.

**ADCC**. Determination of ADCC was performed as described previously[42]. In brief, $CD4^+CCR5^+$ NKR24 target cells that express luciferase under the control of a tat-dependent promoter were infected with replication-competent $SHIV_{SF162P3}$ or $SHIV_{BaL}$ (200 ng/ml p27) by spinoculation at $1200 \times g$ for 2 h with 40 μg/ml polybrene. Three days post spinoculation, $1 \times 10^4$ target cells/well were co-incubated with effector KHYG-1 NK cells at an effector to target ratio of 10:1 with or without serial Ab dilutions in 200 μl assay media (RPMI supplemented with 5 U/ml IL-2) in round bottom 96-well plates at 37 °C and 5% $CO_2$. All Ab and plasma dilutions were plated in duplicate. Effector KHYG-1 NK cells expressing macaque or human CD16 were used for assays with plasma or cloned Ab, respectively. After 8 h co-incubation, each assay well was mixed by pipetting and 150 μl transferred to black flat-bottom plates containing 50 μl Bright-Glo (Promega) and incubated for 2 min at 25 °C. Luminescence was measured on a Victor X Light plate reader (Perkin Elmer) and relative light units (RLU) were normalized according to the following formula: [sample mean – background (mock-infected targets and effectors)] / [maximum (SHIV-infected targets and effectors, no mAb – background)] × 100. ADCC activity is reported as the percentage loss of RLU.

**ADCP**. The phagocytic potential 10E8v4 Fc variants were assessed by a method adapted from Ackerman et al.[64]. Fluorescent antigen beads were prepared by coupling biotinylated MPER peptide to neutravidin conjugated low-intensity yellow fluorescent polystyrene beads (Spherotech Inc., CFL-0852-2), and were confirmed to be properly functionalized by assessing anti-MPER antibody staining alongside negative isotype controls. To measure ADCP, 10E8v4 variants were three-fold serially-diluted to achieve a final concentration range of 3 nM to 4 pM in 96-well tissue culture treated microplates (Corning®, CLS3596), followed by the addition of a homogenous solution of a 1:20 ratio of effector THP-1 monocyte-like cells (ATCC® [TIB-202™]) to target fluorescent antigen beads. Following a 4 h incubation at 37 °C and 5% $CO_2$, the sampled contents of each well were analyzed by flow cytometry, and ADCP results were reported as a phagocytosis score (P-score) metric, defined as the product of the percentage of THP-1 monocytes that engulfed at least one fluorescent bead (the percentage of gated THP-1 cells with FITC signal above a threshold MFI) and the average number of beads engulfed (the MFI signal of the FITC+ gated THP-1 cells).

**Neutralization assays**. Neutralization assays were performed in TZM-bl cells expressing luciferase under the control of the HIV *tat* promoter as standardized by Wei et al. using replication-competent $SHIV_{SF162P3}$ stocks expanded in CD4-enriched rhesus splenocytes or single cycle competent pseudoviruses[65,66]. For pseudovirus assays, three-fold serial dilutions of Ab were incubated in 150 μl with the indicated pseudovirus for 1 h in cDMEM (DMEM [Gibco] supplemented with 4.5 g/l D-glucose, 10% heat-inactivated FBS, 1% L-glutamine, 1% penicillin, 1% streptomycin) at 37 °C and 5% $CO_2$. Next, $1 \times 10^4$ cells/well were added in 50 μl media further supplemented with 7.5 μg/ml DEAE-dextran to aid viral entry. After 48 h co-incubation, 140 μl of media was gently removed from each well and adherent cells were lysed by the addition of 60 μl Bright-Glo (Promega) for 2 min. Eighty microliter was then transferred to black plates and viral infection was determined as the relative level of luciferase activity measured on a Victor X Light plate reader (Perkin Elmer). All experimental samples were plated in duplicate and the percent neutralizing activity was calculated using the following formula: [Mean no Ab (i.e., virus and cells minus cells only background) – Mean sample (i.e., sample value minus cells only background)] / [mean no Ab] × 100. Isotype and positive control Abs were included on all assay runs. Assays with replication-competent virus were performed similarly but read after 24 h instead of 48 h.

**ELISAs**. Assays were performed largely as described by Malherbe et al.[67]. Plates were coated with either: (1) MPER for 10E8v4 Fab binding, (2) anti-10E8v4 idiotype mAb 2D1 for 10E8v4 in plasma or tissues (3) recombinant monomeric $HIV_{SF162}$ gp140 for native autologous plasma responses, or (4) plasma matched 10E8v4 variant for measuring ADA. Flat-bottom plates were coated with the antigen of interest by incubating 0.5–1 μg/ml in 0.2 M $H_2CO_3$ buffer pH 9.4 at 4 °C overnight. Plates were then washed in binding buffer (PBS pH 7.4 + 0.1% Triton X-100) and blocked with 150 μl PBS containing 5% dried milk and 1% goat serum for 1 h at room temperature. Blocking buffer was discarded and three-fold serial dilutions of plasma or mAb were added to unwashed cells in 50 μl binding buffer. After 1 h at room temperature, plates were washed 3× and then incubated for 1 additional hour with 50 μl 1:5000 dilution of goat anti-human H&L (Jackson Laboratories) (assays 1–3) or 50 μl 1:3000 dilution of mouse anti-macaque IgG mAb 1B3 (assay 4) conjugated to horse radish peroxidase (from Invitrogen and NIH AIDS Reagents program, respectively). Plates were then washed 5×, and bound Ab was visualized by the addition of 50 μl tetramethylbenzidine (Southern Biotech) for 10 min before stopping the reaction with 50 μl 1 N $H_2SO_4$. Optical density was immediately quantified on a SoftMax® Pro 5 microplate reader (Molecular Devices) at 450 nm.

**Quantification of plasma and tissue viral load**. Plasma and CAVLs were determined by quantifying SIVgag vRNA from nucleic acid in plasma or PBMCs by quantitative reverse transcription-PCR (RT-PCR) as detailed previously[67]. In brief, 2 μg nucleic acid was amplified for 45 cycles in 30 μl Fast Advanced

Mastermix on a QuantStudio 6 Flex instrument (Applied Biosystems, Life Technologies) and virus copy numbers estimated by comparison to a pBSII-SIVgag standard curve. To measure reservoir virus in tissues from biopsies or necropsy, tissues were homogenized by bead beating. The nucleic acid in tissue homogenates was then analyzed for reservoir virus measured as the SIVgag DNA per μg tissue amplified using ultrasensitive nested quantitative PCR and RT-PCR as detailed previously[67,68]. The following primers and probe were used for all protocols to amplify and detect a conserved region in SIVgag: SGAG21 forward primer (GTCTGCGTCATPTGGTGCATTC), SGAG22 reverse primer (CACTAGKT GTCTCTGCACTATPTGTTTTG), and pSGAG23 probe (5′-6-carboxyfluorescein [FAM]-CTTCPTCAGTKTGTTTCACTTTCTCTTCTGCG-black hole quencher [BHQ1]-3′).

**Immune cell subsets**. Relative composition of immune cell subsets was determined by flow cytometry on freshly thawed cryopreserved PBMCs. Cells from four different rhesus macaques were divided into cluster tubes (Corning™ 4410; $1 \times 10^5$ cells/tube) and washed twice with staining buffer (1 ml phosphate-buffered saline supplemented with 1% FBS and 1 mM EDTA. Live/Dead dye (Fixable Blue, Invitrogen) was added according to the manufacturer's directions and incubated for 15 min at 4 °C protected from light. Cells were then stained for surface markers using the following panel: CD3:BB660, CD4:BUV395, CD8:BUV805, CD14:BV510, CD16:BV711, CD20:PE-Cy594, CD45:PE-Cy7, and HLA-DR:BV650. Cells were then washed twice and analyzed as described above.

**Intracellular cytokine staining**. Specific T cell activity was determined on PBMCs collected at study day 56. PBMCs from each macaque were divided into three polypropylene cluster tubes (Corning™ 4410; $1 \times 10^6$ cells/tube) and washed 2× with cRPMI. Each set of cells was then incubated for 1 h at 37 °C, 5% $CO_2$, and 100% humidity in 125 μl cRPMI supplemented with 3 μg/ml anti-CD28, 3 μg/ml anti-CD49d, and either (A) 0.5% DMSO (mock), (B) 5 μg/ml/peptide of $SIV_{mac239}$ gag pooled peptides (ARP-12364;), or (C) 1 μg/ml ionomycin (Sigma) and 40 pg/ml phorbol 12-myristate 13-acetate (Sigma) as a positive control. Following 1 h incubation, 0.25 μl GolgiStop™ and 0.25 μl GolgiPlug™ (BD Biosciences) in 50 μl cRPMI were added to each tube. Cells were incubated as before for an additional 8 h, then stored at 4 °C for up to 10 h prior staining. For analysis, cells were washed twice with staining buffer then incubated for 15 min at 4 °C protected from light with Live/Dead dye (Fixable Blue, Invitrogen). Surface markers were then stained using the following panel: CD3:BV510, CD4:BUV395, CD8:BUV805, CD45RA:PE-Cy7, CD95:BV786, CD107a:FITC. Following fixation in 1% PFA (20 min at 4 °C and permeabilization with 0.5% saponin in staining buffer, cells were stained for intracellular markers as follows: CD69:ACP-H7, IL-2:PE, IL-4:BV421, IL-21:APC, IFNγ:BV711, and TNFα:BV650. Stained cells were analyzed on a FACSymphony A5 (BD Biosciences) and data were analyzed with FlowJo software.

**Splenocyte infection assays**. Cryopreserved macaque splenocytes were washed twice with RPMI, then resuspended at a concentration of $1 \times 10^6$ cells/ml in RPMI supplemented with 4.5 g/l D-glucose, 10% heat-inactivated FBS, 1% L-glutamine, 1% penicillin, and 1% streptomycin (cRPMI) plus either 300 U/ml recombinant rhesus IL-4 and 150 U/ml granulocyte-macrophage colony-stimulating factor to enrich macrophage-derived dendritic cells or with 10 μg/ml phytohemagglutinin to expand T cells. Stimulated splenocytes were incubated for 4 days at 37 °C with 5% $CO_2$ and 100% humidity and then refreshed with new cRPMI plus growth factors. Seven days after stimulation, T cell and MDC-enriched splenocytes were collected for assay setup, using a 7 min incubation with TrypLE Express (Fisher) to dissociate adherent cells from flasks. Cells were washed twice with serum-free cRPMI (no FBS), resuspended at $2 \times 10^6$ cell/ml in cRPMI-no FBS, and 0.5 ml was added to each well of a 24-well culture plate. The indicated mAb was diluted in cRPMI-no FBS to 100 μg/ml and 50 μl mAb then added to the appropriate well for 30 min to pre-load cells with mAb. Separately, $SHIV_{SF162P3}$ virus stock (p27 concentration of 788 ng/ml) was incubated with 22.2% either heat-inactivated or normal macaque serum pooled from ten macaques for 30 min at room temperature using a ratio of 350 μl virus and 100 μl serum per assay well. After separate cell-mAb and virus-serum incubations, 450 μl of the virus-serum mixture was added to the appropriate well, giving a final assay concentration of $1 \times 10^6$ cells/5 μg mAb/275 ng p27 virus/10% heat-inactivated or normal macaque serum/ml/well. Splenocyte cultures were then incubated as above and infection monitored longitudinally by flow cytometry staining for viability (Live/Dead Fixable Yellow; Fisher), surface markers CD3:PB (BD-558124), CD4:APC (Miltenyi-130-091-232), CD11b/CD18:PE-Cy7 (eBioscience-25-0118-42), and CD11c:PerCP (Invitrogen-MA1-10087), and intracellular p27:FITC (ARP-1610). All test conditions were plated in duplicate and similar results were obtained from repeating the experiment two additional times with splenocytes from different macaques.

**Statistical analysis**. Statistical analysis was performed as indicated in the figure legends. Ordinary one-way ANOVAs followed by Tukey's post-hoc test for multiple comparisons were used to analyze data in Figs. 4b, c and 5b, c, g, h, followed by Dunnett's post-hoc test for comparison to control group in Figs. 4f, h, 5g, and 7e, f. Two-way ANOVA was used to analyze data in Figs. 4i–l and 5h followed by Tukey's post-hoc test for multiple comparisons. Differences in individual tissues in

Fig. 4m were analyzed by two-way repeated-measures ANOVA followed by Tukey's post-hoc test for multiple comparisons. Correlation plots in Figs. 6b, d, f and 7e, f were analyzed by one-tailed Spearman tests and visualized with a simple linear regression line. Viral load is presented and analyzed using $\log_{10}$ transformed data. Statistical significance was determined at the significant alpha level of 0.05. All statistical analysis was performed in GraphPad Prism 9.

**Reporting summary**. Further information on research design is available in the Nature Research Reporting Summary linked to this article.

## Data availability

All datasets analyzed during this study are included in this published manuscript and its Supplementary Information. Source data for all figures and supplementary materials are provided with this paper. All other data are available from the corresponding author on reasonable request. Source Data are provided with this paper.

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

## Acknowledgements
We thank Diane Kubitz and Jonathan Otsuji at The Scripps Research Institute Antibody Production Core facility for antibody production, Christina Corbaci for assistance with figure design and production, David Evans for providing the CD4+CCR5+ NKR24 and KHYG-1 NK cell lines, and the NIH/NIAID Vaccine Research Center for the expression plasmid for 10E8v4 and the anti-10E8 mAb 2D1. We acknowledge the NIH AIDS Reagent Program for providing the following reagents: anti-HIV-1 Env mAb b12 (ARP-2640), TZM-bl cells with FcRs (ARP-11796, ARP-11797, ARP-11798), SIVmac239 gag pooled peptides (ARP-12364), and SHIVSF162P3 virus stock (ARP-6526). This work was supported by the following grants: R01 AI129801 (A.J.H.), P51 OD011092, and U42 OD023038-03.

## Author contributions
A.J.H. conceived the study, obtained funding, designed the experiments, analyzed the data, and supervised the research. D.A.S. co-managed the study, performed all flow cytometry experiments, and analyzed the data. O.S. and N.L.H. contributed valued advice; B.S.G. and M.E.A. developed assays, tested Fc variant mAbs, and provided data analysis; D.A.S., B.S.G., S.P., T.C., J.D., P.B., W.F.S., H.H., and T.B. designed and performed in vitro experiments; S.P. managed animal selection and assignments, supervised in vitro work, managed tissue collections and specimen inventory. D.A.S., R.A., and L.G. performed statistical analysis; D.A.S. and A.J.H. wrote and edited the manuscript with assistance from coauthors.

## Competing interests
The authors declare no competing interests.
