## [Peer Review File · Nature Communications]

REVIEWER COMMENTS

Reviewer #1 (Remarks to the Author):

NCOMMS-21-29317

A broadly neutralizing antibody (bNAb) to a membrane-proximal external region (MPER) epitope of HIV-1 gp41 (10E8v4) and variants of 10E8v4 with Fc domain substitutions that enhance binding to complement C1q and Fcγ receptors (FcγRII & FcγRIII) (EFTAE) or impair FcRII/III binding (LALA) were administered at sub-protective doses (5 mg/kg) to separate groups of six rhesus macaques. Three days later, the animals were challenged intrarectally with SHIV-BalP4 (500 TCID₅₀). All of the animals became infected and peak viral loads for the three antibody treated groups did not differ relative to the untreated control group. Viral loads during the post-acute phase of infection also did not differ for the groups that received the Fc variants of 10E8v4. However, post-acute viral loads for the group that received unmodified 10E8v4 were reduced 0.14-fold relative to the control group ($p=0.0012$). A subsequent challenge study in which the EFTAE variant and unmodified 10E8v4 were given to macaques at higher doses was performed to determine if higher concentrations of 10E8v4 EFTAE could also reduce post-acute viral loads. In this case, despite lower concentrations of the EFTAE variant in plasma compared to unmodified 10E8v4, post-acute viral loads in both groups were significantly reduced. Based on these results, the authors conclude that the observed reductions in post-acute viral loads are dependent on intact Fc-mediated functions. At the time of challenge (TOC), neutralization titers for each group did not differ (at least for the first study) and neither antibody-dependent complement deposition (ADCP) nor antibody-dependent complement-mediated lysis (ADCML) correlated with post-acute viral loads. However, antibody-dependent cellular phagocytosis (ADCP) measured using MPER peptide-coated beads correlated inversely with post-acute viral loads ($p=0.0163$). A role for antibody-dependent cellular cytotoxicity (ADCC) was excluded because 10E8v4 (unmodified or EFTAE) was unable to mediate ADCC against SHIV-BalP4-infected cells. There were no obvious differences between antibody treated groups in endogenous virus-specific immune responses. From this data, the authors conclude that their results “demonstrate that phagocytic effector functions can contribute to bNAb efficacy against neutralization insensitive viruses”.

This is an impressive manuscript for its carefully designed animal studies and thorough analysis of antibody effector functions. The quality of the data is generally high and the manuscript is well written. Nevertheless, the data does not support the greatly overstated conclusions of the paper. Contrary to the title and statements in the abstract (and elsewhere), the data are merely correlative and do not definitively show that ADCP reduces post-acute viremia. The observed reductions in post-acute viral loads are modest (~1 log) and the only evidence in support of a phagocytic effector function comes from a similarly modest inverse correlation with ADCP measured using peptide-coated beads.

Peptide-coated beads do not represent the configuration of the MPER epitope on virus particles and infected cells. A plausible explanation for the inability of 10E8v4 to mediate ADCC is that when this antibody is bound to Env on infected cells, its Fc domain is oriented close to the cellular membrane and is not accessible for FcR interactions. If similar mechanisms impair the recognition of 10E8v4 on virions and infected cells by FcRs on phagocytic cells, assays using peptide-coated beads could grossly overestimate ADCP.

It is unclear how transient levels of 10E8v4 for a few weeks after antibody transfer could result in long-term reductions in chronic phase viral loads. Although endogenous immune responses were investigated, these analyses appear to be limited to binding antibody titers and to the measurement of Env-specific T cell responses at a single time point. To exclude a role for endogenous immune responses, these analyses should be expanded to compare neutralizing antibody titers and T cell responses to multiple viral antigens at additional time points.

Reviewer #2 (Remarks to the Author):

Manuscript from Spencer et al. decipher the contribution of complement activity of bNAbs using the experimental SHIV infection NHP model. For this study, they select a bNAbs found to be insensitive to ADCC but highly efficient in Complement lysis of SHIVSF162 virions using several in vitro assays. They further produced bNAb mutants by selected point mutations in order to abrogate or enhance FcR binding and/or interaction with C.

Analysis of NHP protection from SHIV experimental challenge was performed using sub-neutralizing doses of Abs for improving the detection of the protective effect related to C-mediated effector functions. Unfortunately, results showed that enhancing C effector functions did not lead to increased SHIV protection, therefore, invalidating their hypothesis and the construction of the project. Judiciously, authors catches up on ADCP at TOC to explain the post-acute decreased viral load.

This study is a very thoughtful and intelligent study taking into account the numerous caveats and limitations already largely described for such protective experimental SHIV study. The ADCP Fc-mediated functional activity proposed here to participate in viral load decrease is of possible relevant interest.

The overall results shown rise a few question that may be clarified by the authors

- What is the ADCP activity of the mAb and their mutant? their IC50? similarly to what was studied for ADCC and ADCML in figure 1.

- Can the ADCML and ADCC assays used in these studied be considered as relevant for in vivo functions? Many different in vitro assays have been developed for studying Fc-mediated function. These assays did not necessarily give similar results. They are not yet validate, standardized, and associated to effective in vivo functions. Authors should therefore limit their conclusions to the in vitro assay they performed in this study. For example line 90 “but no ADCC activity” to “but no ADCC activity in the in vitro assay using NKR24 target cells and KHYG-1 NK effector cells”. This limitation may also be discussed in the discussion section.

Additional comments

As Ab distribution in the blood is lower for EFTAE mutant (Figure 4A), it would have been of interest to compare virus load evolution with similar blood Ab concentration. For example comparing virus load for EFTAE mutant at 10-20 mg (Figure 5) with that of unmodified or LALA Ab at 5mg (figure 4). Would the Post-acute PVL be different?

Point-by-point responses to REVIEWER COMMENTS

NCOMMS-21-29317A

Reviewer #1 (Remarks to the Author):

We have itemized our responses to each reviewer's concerns below (*italicized bold type*).

A broadly neutralizing antibody (bNAb) to a membrane-proximal external region (MPER) epitope of HIV-1 gp41 (10E8v4) and variants of 10E8v4 with Fc domain substitutions that enhance binding to complement C1q and Fc γ receptors (Fc γ RII & Fc γ RIII) (EFTAE) or impair FcRII/III binding (LALA) were administered at sub-protective doses (5 mg/kg) to separate groups of six rhesus macaques. Three days later, the animals were challenged intrarectally with SHIV-BaIP4 (500 TCID₅₀). All of the animals became infected and peak viral loads for the three antibody treated groups did not differ relative to the untreated control group. Viral loads during the post-acute phase of infection also did not differ for the groups that received the Fc variants of 10E8v4. However, post-acute viral loads for the group that received unmodified 10E8v4 were reduced 0.14-fold relative to the control group (p=0.0012). A subsequent challenge study in which the EFTAE variant and unmodified 10E8v4 were given to macaques at higher doses was performed to determine if higher concentrations of 10E8v4 EFTAE could also reduce post-acute viral loads. In this case, despite lower concentrations of the EFTAE variant in plasma compared to unmodified 10E8v4, post-acute viral loads in both groups were significantly reduced. Based on these results, the authors conclude that the observed reductions in post-acute viral loads are dependent on intact Fc-mediated functions. At the time of challenge (TOC), neutralization titers for each group did not differ (at least for the first study) and neither antibody-dependent complement deposition (ADCP) nor antibody-dependent complement-mediated lysis (ADCML) correlated with post-acute viral loads. However, antibody-dependent cellular phagocytosis (ADCP) measured using MPER peptide-coated beads correlated inversely with post-acute viral loads (p=0.0163). A role for antibody-dependent cellular cytotoxicity (ADCC) was excluded because 10E8v4 (unmodified or EFTAE) was unable to mediate ADCC against SHIV-BaIP4-infected cells. There were no obvious differences between antibody treated groups in endogenous virus-specific immune responses. From this data, the authors conclude that their results “demonstrate that phagocytic effector functions can contribute to bNAb efficacy against neutralization insensitive viruses”.

We thank the reviewer for the thoughtful comments and suggestions for manuscript revisions. We appreciate the reviewer's complementary remarks regarding the study design, manuscript preparation, high data quality and analysis related to effector functions.

Please note that the challenge virus was SHIV-SF162P3 (300 TCID₅₀ as measured in rhesus PBMC), a neutralization resistant, Tier 2, virus. Reviewer inadvertently stated that SHIV-BaL.P4 is a Tier 1 virus was the challenge virus in our study. It is important to clarify the challenge virus because the supporting in vitro data with SHIV-SF162P3 is most relevant in our principal observations and conclusions.

Please also note that HIV-BaL was used when screening a large panel of bNAbs (Fig.1A) with diverse specificities and measurable lysis activities in an effort to make controlled but workable comparisons. This data provides a confirmation and control experiment showing that unmodified 10E8v4 mediates viral lysis, and therefore is an appropriate bNAb to study as a lysis enhanced variant as well as a lysis ablated variant.

The Reviewer highlights our conclusion, “From this data, the authors conclude that their results “demonstrate that phagocytic effector functions can contribute to bNAb efficacy against neutralization insensitive viruses””. Please note our response below to Reviewer #2 in support of our conclusion.

This is an impressive manuscript for its carefully designed animal studies and thorough analysis of antibody effector functions. The quality of the data is generally high and the manuscript is well written. Nevertheless, the data does not support the greatly overstated conclusions of the paper. Contrary to the title and statements in the abstract (and elsewhere), the data are merely correlative and do not definitively show that ADCP reduces post-acute viremia.

We appreciate these positive comments and have modified the title, abstract, and manuscript to emphasize that our conclusion of the inherent phagocytic activity in unmodified 10E8v4 is associated with the reduced post- acute viremia. Language asserting phagocytosis as the primary mechanistic factor has been removed.

Additionally, we have revised the discussion to incorporate the nuances of this conclusion, highlighting the work performed that was unsuccessful at identifying other possible mechanisms to reduce viral load, and suggesting a possible mechanism for ADCP to mediate this effect.

Our conclusion of the most likely mechanism of ADCP playing a significant role in reducing post viremia is based on a number of in vitro outcomes in this study and also supported by in vivo outcomes not attributable to neutralization with MPER-targeting Abs (vaccine-induced and passive mAbs) in several NHP studies (referenced in the manuscript). Respectfully, we also point the reviewer to the recent Pegu et al (2019, PNAS) meta analysis of passive studies and the relationship between serum-neutralizing antibody titer and protection against SHIV challenge. A key finding of MPER targeting bNAbs is the surprising but consistent outcome of protection with MPER bNAbs at serum titers below an expected threshold of neutralization expected for protection. A clear explanation is still being debated, but other recent revelations when assessing antibody activity during viral fusion events include a study this year giving structural and functional insights into antibody activity during viral fusion (Montefiori 2021,PNAS). A commentary by Burton (2021, PNAS) expanded on a potential mechanism of fusion inhibitor -like MPER bNAbs, such as 10E8v4, and suggests that during early events of infection in NHP passive studies, i.e., after the initial

contact of virus with target cells, MPER becomes exposed setting up a scenario where MPER bNAbs bound monomerically to high-affinity Fc γ RI on phagocytes or dendritic cell targets provide a protection advantage mediated by phagocytic activity. Thus, MPER bNAbs are particularly noted for their unexpected ability to protect against SHIV challenges even when present at low serum concentrations, making MPER bNAbs an especially interesting group to define the precise mechanism(s) of protection.

The observed reductions in post-acute viral loads are modest (~1 log) and the only evidence in support of a phagocytic effector function comes from a similarly modest inverse correlation with ADCP measured using peptide-coated beads.

While the reduction in viremia is modest, it was highly reproducible and consistent between groups where high levels of antibody with intact effector function was present (5 mg/kg unmodified (Fig. 4), 10+ mg/kg unmodified and 10+ mg/kg EFTAE (Fig. 5). No reduction was observed in the group without effector functions (5 mg/kg LALA) or when antibody was rapidly cleared (5 mg/kg EFTAE, Fig. 4). Overall, there were 18 animals with no or little expected effector activity (6 controls, 6 LALA, 6 low dose rapidly cleared EFTAE) which showed similar post-acute viremia and 14 animals with effector functions present (6 low dose unmodified, 4 high dose unmodified, and 4 high dose EFTAE) which had the modest but consistent reduction in post-acute viral load.

The modest reduction is likely a maximum effect in this experimental setting, as supported by the similar reduction observed with higher doses of antibody (Fig. 5), and this is consistent with other literature showing modest effects of effector functions (0-45% as cited in the manuscript).

Peptide-coated beads do not represent the configuration of the MPER epitope on virus particles and infected cells. A plausible explanation for the inability of 10E8v4 to mediate ADCC is that when this antibody is bound to Env on infected cells, its Fc domain is oriented close to the cellular membrane and is not accessible for FcR interactions.

In the case of infected cells, (as opposed to pre-fusion events as discussed above), ADCC mediated by the low-affinity Fc γ RIII, could be limited due to steric hindrance. This and other considerations are included in the second paragraph of the discussion, which highlights the need for effector functions for each bNAb to be measured independently against each virus of interest, and in the fourth paragraph, the limitations of in vitro assays. It is important to note here that while 10E8v4 IgG1 has no ADCC activity in this cell based assay against SHIV_{SF162P3} and other tier 2 viruses, it does have some ADCC activity in this cell based assay against other HIV strains (Alpert et al 2012, J. Virology and Duchemin et al 2018, Frontiers Immunology).

If similar mechanisms impair the recognition of 10E8v4 on virions and infected cells by FcRs on phagocytic cells, assays using peptide-coated beads could grossly overestimate ADCP.

We thank the reviewer for highlighting this risk and agree that the absolute value of ADCP activity and assay signal to noise may be greater in the bead based assay based on previous comparisons between the bead based and virion-based phagocytosis assays (Tay et al 2016, PLoS Path), however results have been well-correlated between the two (Chu et al 2020, PLoS Path). The bead based assay has been widely published as a standard metric for in vivo ADCP activity (Hangartner et al 2021, Sci Trans Med; Asokan et al 2020, PNAS; Om et al 2020, PNAS, and importantly, has also correlated with reduced risk of infection in humans (Neidich et al 2019, J Clin Invest). We believe that in this study, the relative degree of ADCP activity between groups is accurately reflected by the bead based assay

It is unclear how transient levels of 10E8v4 for a few weeks after antibody transfer could result in long-term reductions in chronic phase viral loads.

We believe a mechanistic investigation into this phenomenon is warranted for MPER bnAbs, albeit outside the scope of this study. Nonetheless, we have expanded the third paragraph of the discussion to propose a conceivable mechanism. In short, we hypothesize that early seeding events in mucosal and lymph node sites influence downstream pathogenesis. We base this hypothesis not only on observations from this study but also on other investigations in our and other laboratories showing a consequence of viral dosing and initial viral seeding. We have evidence from pre-exposure studies with different doses of SHIV combined with subsequent mAb treatments of varying potency, efficacy, or function, the impact of early viral seeding is evident and discernable in the eventual in vivo viremia and pathogenic outcomes. On this point and, we believe relevant here, we concluded in Shapiro et al Nat Comm 2020: "Understanding the longevity of the earliest seeding events and kinetics of persistent reservoir establishment is critical to elucidating the mechanism of tight control after early intervention."

Although endogenous immune responses were investigated, these analyses appear to be limited to binding antibody titers and to the measurement of Env-specific T cell responses at a single time point. To exclude a role for endogenous immune responses, these analyses should be expanded to compare neutralizing antibody titers and T cell responses to multiple viral antigens at additional time points.

We have added an additional figure (7D) to show the lack of neutralizing activity in plasma among all groups and at all time points measured against the replication competent challenge virus. We have expanded the T cell analysis to report the average T cell response from each animal from 3 timepoints, Day 28, Day 42, and Day 56. (Note in some cases, sample availability was restricted and

alternative timepoints were used. For a few animals only 1-2 timepoints were able to be collected. The raw data detailing each average value reported in Figure 7 is provided in the excel data table.) The original Day 56 timepoint was performed against a complete GAG peptide pool and the additional two timepoints were against complete GAG + Env Potential T Cell Epitopes peptide pools to detect T cells to multiple viral antigens.

Reviewer #2 (Remarks to the Author):

Manuscript from Spencer et al. decipher the contribution of complement activity of bNAbs using the experimental SHIV infection NHP model. For this study, they select a bNAbs found to be insensitive to ADCC but highly efficient in Complement lysis of SHIVSF162 virions using several in vitro assays. They further produced bNAb mutants by selected point mutations in order to abrogate or enhance FcR binding and/or interaction with C. Analysis of NHP protection from SHIV experimental challenge was performed using sub-neutralizing doses of Abs for improving the detection of the protective effect related to C-mediated effector functions. Unfortunately, results showed that enhancing C effector functions did not lead to increased SHIV protection, therefore, invalidating their hypothesis and the construction of the project. Judiciously, authors catch up on ADCP at TOC to explain the post-acute decreased viral load.

We thank the Reviewer for the thoughtful comments and suggestions for revisions. We appreciate the reviewer's comments regarding the study design to account for the nuances of these SHIV protection studies. We have itemized our responses to each concern below.

This study is a very thoughtful and intelligent study taking into account the numerous caveats and limitations already largely described for such protective experimental SHIV study. The ADCP Fc-mediated functional activity proposed here to participate in viral load decrease is of possible relevant interest. The overall results shown raise a few questions that may be clarified by the authors:

-What is the ADCP activity of the mAb and their mutant? their IC50? similarly to what was studied for ADCC and ADCML in figure 1.

We appreciate the Reviewer noting that this data should be included. In response, we respectfully point to Figure 2D, and trust that this should provide the Reviewer with this information.

-Can the ADCML and ADCC assays used in these studied be considered as relevant for in vivo functions? Many different in vitro assays have been developed for studying Fc-mediated function. These assays did not necessarily give similar results. They are not yet validate, standardized, and associated to effective in vivo functions.

We thank the reviewer for highlighting this important point. The ADCC assay has been widely used as the standard in the field since its development in 2012 by Alpert et al, Journal of Virology, with a slight modification to the target cell line to

better accommodate susceptibility to some HIV strains by Thomas et al 2020, J. Immunological Methods. Please see recent peer-reviewed studies, e.g. used in Hangartner et al 2021, Sci Trans Med; Asokan et al 2020, PNAS. Like any in vitro assay, direct relevance to activities in vivo is difficult to establish, although the relationships between ADCC and reduced risk of infection in RV144 and various NHP studies suggest relevance. ADCML, indeed, complement-mediated activities in general, have been less well-studied, which was a gap we sought to address in this study. In the revised manuscript, we now highlight the many assays available, their differing results, and the open questions about their direct biological relevance in vivo.

Authors should therefore limit their conclusions to the in vitro assay they performed in this study. For example, line 90 “but no ADCC activity” to “but no ADCC activity in the in vitro assay using NKR24 target cells and KHYG-1 NK effector cells”. This limitation may also be discussed in the discussion section.

We have modified the text throughout the manuscript to reflect this important point. Additionally, we have added text to the fourth paragraph of the discussion to highlight this limitation of the manuscript.

Additional comments

As Ab distribution in the blood is lower for EFTAE mutant (Figure 4A), it would have been of interest to compare virus load evolution with similar blood Ab concentration. For example, comparing virus load for EFTAE mutant at 10-20 mg (Figure 5) with that of unmodified or LALA Ab at 5mg (figure 4). Would the Post-acute PVL be different?

This is a good point and was done when analyzing the data. Yes, the post-acute PVL is significantly lower in the groups with both effector functions intact and antibody present at comparable levels. We did not present this analysis in the manuscript to avoid recycling data, but it has now been added as Figure S3, and includes the p values for all comparisons against the no effector function groups (control and LALA).

REVIEWERS' COMMENTS

Reviewer #1 (Remarks to the Author):

The interpretation of the revised manuscript is less overstated. However, the connection between post-acute viral loads in animals and in vitro measures of ADCP remains tenuous. The role of ADCP is supported only by a moderately significant inverse correlation between post-challenge viral loads and ADCP measured at single time point with an assay of dubious physiological relevance. With so many previous publications reporting twilight associations between different Fc-mediated antibody functions and various outcomes of challenge in NHPs (of which some are selectively cited), how does this study advance the field?

Reviewer #2 (Remarks to the Author):

The manuscript was thoroughly revised, nicely addressing the reviewer's comments. According to this new version of the manuscript, in vitro assays performed show that 10E8 exhibit ADCP but not ADCML or ADCC on SHIVSF162. However, authors stipulate line 94-96 on figure 1 A, B that 10E8 show comparatively high levels of ADCML but no ADCC activity and that 10E8 was therefore selected for C' and phagocytic effector function. These comments are misleading as 1) ADCML was not detected against virus SHIV SF162. ADCML and ADCC should be address on the same virus SHIVSF162 (Figure 1I and 1B) and 2) phagocytic effector function mentioned line 96 are only analyzed line 127 and figure 2D. Manuscript would increase in clarity if author correct these points.

Point-by-point responses to REVIEWER COMMENTS – 2nd review

NCOMMS-21-29317B

Our responses to each reviewer's concerns are shown below in *italicized* type.

Reviewer #1 (Remarks to the Author):

The interpretation of the revised manuscript is less overstated. However, the connection between post-acute viral loads in animals and in vitro measures of ADCP remains tenuous. The role of ADCP is supported only by a moderately significant inverse correlation between post-challenge viral loads and ADCP measured at single time point with an assay of dubious physiological relevance. With so many previous publications reporting twilight associations between different Fc-mediated antibody functions and various outcomes of challenge in NHPs (of which some are selectively cited), how does this study advance the field?

Beyond further support for contributions of antibody-mediated effector functions to protect from infection, we find the additional advance to the field is that this study is the first meaningful in vivo test of complement-enhanced activity of Fc variants.

We appreciate the reviewer's recognition of the limitations of the study, which we have now worked to ensure are made sufficiently clear in the present manuscript version. Whether it changes the reviewer's opinion of the bead-based ADCP assay or not, respectfully, we would reiterate that direct phagocytosis of virions is expected to have potential physiologic relevance, and recently published findings demonstrate closely correlated results between the bead-based and virion-based phagocytosis assays (Tay et al 2016, PLoS Path).

As we have proposed in the discussion section, in the absence of meaningful neutralization activity or differences between groups in C' deposition or viral lysis activity at the time of challenge, phagocytic activity induced by antibody-antigen complexes is, in our opinion, a reasonable effector activity to account for reduced viral seeding and subsequent downstream viral load reduction.

We would be happy to further round out appropriate citations if the reviewer is able to make their identity clear. As we have cited excellent work in which effector functions did not contribute to protection in prior passive protection experiments in NHP, we must admit to being at a loss to identify the specific literature the reviewer alludes to.

Lastly, for single challenge passive protection study protocols, the most significant timepoint to evaluate the activity of the onboarded antibody is at the time of challenge. It is unclear to us why or what additional timepoints during infection would be informative regarding the antibody being studied.

Reviewer #2 (Remarks to the Author):

The manuscript was thoroughly revised, nicely addressing the reviewer's comments. According to this new version of the manuscript, in vitro assays performed show that 10E8 exhibit ADCP but not ADCML or ADCC on SHIVSF162. However, authors stipulate line 94-96 on figure 1 A, B that 10E8 show comparatively high levels of ADCML but no ADCC activity and that 10E8 was therefore selected for C' and phagocytic effector function. These comments are misleading as 1) ADCML was not detected against virus SHIV SF162. ADCML and ADCC should be address on the same virus SHIVSF162 (Figure 1I and 1B) and 2) phagocytic effector function mentioned line

96 are only analyzed line 127 and figure 2D. Manuscript would increase in clarity if author correct these points.

We appreciate the acknowledgement by the reviewer that we adequately addressed the initial comments. In further response to the new comments, we thank the reviewer for pointing out that we should not include the phrase “and phagocytic effector functions” when describing our motivation to choose 10E8v4 as the bNAb to evaluate antibody-mediated C' lysis using the EFTAE variant. That phrase has been removed.

Of the in vitro data shown, the selection of 10E8v4 for this study was indeed based on activities observed on different HIV-1 strains. For all data directly related to the in vivo outcomes, please note that both ADCML and ADCC against the SHIV challenge virus is shown: Fig. 1b, 2e (ADCC), Fig. 1i (ADCML). We also show complete data sets representing all animal groups for ADCML against the SHIV challenge virus at the time of challenge in Figure 6e. In our initial assessment of bNAbs, we used HIV_{BaL} virus to generate ADCML data. This work was done before the macaque protection study and illustrates a range of ADCML activity in a bNAb panel (Fig. 1a).